# Protein context shapes the specificity of SH3 domain-mediated interactions in vivo

Ugo Dionne[1,2,3,4,5], Émilie Bourgault[3,4,5,6,13], Alexandre K. Dubé [3,4,5,6,7,13], David Bradley[3,4,5,6,7,13], François J. M. Chartier [1,2,3,13], Rohan Dandage[3,4,5,6,7], Soham Dibyachintan[3,4,6,7,8], Philippe C. Després [3,4,5,6], Gerald D. Gish[9], N. T. Hang Pham[3,10], Myriam Létourneau[3,10], Jean-Philippe Lambert [1,2,11], Nicolas Doucet[3,10], Nicolas Bisson [1,2,3,12✉] & Christian R. Landry [3,4,5,6,7✉]

Protein–protein interactions (PPIs) between modular binding domains and their target peptide motifs are thought to largely depend on the intrinsic binding specificities of the domains. The large family of SRC Homology 3 (SH3) domains contribute to cellular processes via their ability to support such PPIs. While the intrinsic binding specificities of SH3 domains have been studied in vitro, whether each domain is necessary and sufficient to define PPI specificity in vivo is largely unknown. Here, by combining deletion, mutation, swapping and shuffling of SH3 domains and measurements of their impact on protein interactions in yeast, we find that most SH3s do not dictate PPI specificity independently from their host protein in vivo. We show that the identity of the host protein and the position of the SH3 domains within their host are critical for PPI specificity, for cellular functions and for key biophysical processes such as phase separation. Our work demonstrates the importance of the interplay between a modular PPI domain such as SH3 and its host protein in establishing specificity to wire PPI networks. These findings will aid understanding how protein networks are rewired during evolution and in the context of mutation-driven diseases such as cancer.

[1] Centre de Recherche du Centre Hospitalier Universitaire (CHU) de Québec-Université Laval, Québec, QC, Canada. [2] Centre de Recherche sur le Cancer de l'Université Laval, Québec, QC, Canada. [3] PROTEO-Regroupement Québécois de Recherche sur la Fonction, l'Ingénierie et les Applications des Protéines, Québec, QC, Canada. [4] Centre de Recherche en Données Massives de l'Université Laval, Université Laval, Québec, QC, Canada. [5] Institut de Biologie Intégrative et des Systèmes (IBIS), Université Laval, Québec, QC, Canada. [6] Département de Biochimie, de Microbiologie et de Bio-Informatique, Université Laval, Québec, QC, Canada. [7] Département de Biologie, Université Laval, Québec, QC, Canada. [8] Department of Chemical Engineering, Indian Institute of Technology Bombay (IIT), Powai, Mumbai, Maharashtra, India. [9] Lunenfeld-Tanenbaum Research Institute, Mount Sinai Hospital, Joseph and Wolf Lebovic Health Complex, Toronto, ON, Canada. [10] Centre Armand-Frappier Santé Biotechnologie, Institut national de la recherche scientifique (INRS), Université du Québec, Laval, QC, Canada. [11] Département de Médecine Moléculaire, Université Laval, Québec, QC, Canada. [12] Département de Biologie Moléculaire, Biochimie Médicale et Pathologie, Université Laval, Québec, QC, Canada. [13] These authors contributed equally: Émilie Bourgault, Alexandre K. Dubé, David Bradley, François J.M. Chartier. ✉email: Nick.Bisson@crchudequebec.ulaval.ca; christian.landry@bio.ulaval.ca

Proteins often display a modular architecture defined by folded domains that bind short linear peptide motifs on their interaction partners[1]. Modular domains are generally considered to act as "beads on a string" by virtue of their ability to independently fold and bind target peptides with high intrinsic specificity in vitro[2–5]. However, binding domains are often part of larger proteins that can comprise many functional elements. Whether/how PPI domain binding specificity is modulated by positioning within their host protein and/or intramolecular interactions (collectively defined here as protein context) remains poorly defined. Such regulation of interaction specificity would imply that during evolution and in disease states, mutations occurring either within or outside a modular binding domain could alter its protein interaction specificity. We examined this question by studying PPIs of proteins containing SRC Homology 3 domains (SH3s) in vivo. SH3s are one of the most prevalent families of modular binding domains, having expanded in number throughout evolution with 27 in yeast (on 23 proteins) and nearly 300 in human[6–9]. These ~60 amino acid domains are present on signaling proteins, regulating functions such as endocytosis and actin cytoskeleton remodeling[8]. SH3s typically bind to Pro/Arg-rich peptide motifs on their target partners with an archetypical PXXP motif (where X represents any amino acid)[6,10].

In this work, we combine genome editing, cellular phenotyping, and proteomics to determine the in vivo contribution of protein context to SH3 domain specificity and functions. We find that these PPI modules rarely mediate interactions independently and that their position within their host is important for biological processes such as endocytosis and phase separation. Our results contribute to the current understanding of how PPI networks achieve specificity and how it is altered by mutations, domain gains, and losses.

## Results and discussion

**SH3s contribute to PPI networks complexity and protein function in vivo.** To assess the requirement of SH3 domains for PPIs in vivo, we first measured binary interactions between 22 WT SH3-containing proteins (Supplementary Fig. 1A) as baits and 575 putative partners from their interconnected signaling networks using the dihydrofolate reductase protein–fragment complementation assay (DHFR-PCA) in the budding yeast *Saccharomyces cerevisiae*. About 33% (202/607) of the PPIs detected were described before, mostly by direct methods (Supplementary Fig. 1B, C)[11]. We repeated the experiments with baits in which the SH3s were individually replaced with a flexible linker by genome editing (SH3 domain deletion/stuffing, Fig. 1A). The stuffer DNA sequence (GGCGGAAGTTCTGGAGGTGGTGGT) codes for a small flexible poly-Gly with Ser peptide (GGSSGGGG) that is inspired by previous experiments[12,13]. About a third of the SH3-containing protein interactome is qualitatively or quantitatively SH3-dependent (171 PPIs out of 607, Fig. 1B and Supplementary Fig. 1B, C). We validated the vast majority of the quantitative changes in low-throughput experiments and excluded that this is simply due to changes in protein abundance in most cases (Supplementary Fig. 1D–G). SH3-binding motifs[14] are more frequent in the SH3-dependent PPI partners when compared to sequences from random proteins, SH3-independent PPI partners or PPI partners that are stronger or gained following SH3 deletions (p values = $1.5 \times 10^{-17}$, 0.0038 and $2.1 \times 10^{-08}$, respectively; Mann–Whitney test, one-tailed, Fig. 1C). The changes we measure are therefore enriched for PPIs that depend on the direct interactions of the domains with the PPI partners. Interestingly, the enrichment of SH3 motifs among SH3-dependent PPI partners does not change significantly (p = 0.25, Mann–Whitney test, two-tailed, Fig. 1D) when assigning an

SH3-binding motif randomly to a set of interactors (for example, when testing the enrichment of Protein X preferred SH3 motif among Protein Y SH3-dependent PPI partners). This indicates that SH3-binding motifs determined in vitro can adequately identify SH3-dependent interactions but not discriminate which partner binds to which SH3 domains in vivo.

Surprisingly, 37 PPIs are increased and 75 are gained following SH3 deletions (Fig. 1B). Some of these PPIs have previously been detected with other methods (Supplementary Fig. 1B, C). It is therefore likely that SH3 deletion alters protein folding or positioning within a complex or its relative binding preference, making these gained PPIs now detectable by DHFR-PCA. This observation may also be explained by changes in intramolecular interactions, as reported for the human SH3-containing SRC-family of tyrosine kinases[15].

We next assessed the contribution of SH3s to cellular phenotypes by measuring the growth of WT, SH3-deleted, and knockout strains under 51 different stress conditions ranging from DNA damage induction to high osmolarity (Fig. 1E, F). In most cases, knockout or SH3 deletion leads to subtle phenotypes (Fig. 1E), as expected for nonessential genes[16]. When full gene deletion results in strong phenotypes, SH3 deletion consistently leads to similar growth defects (e.g., *NBP2*, *BBC1*, *BEM1*, and *SLA1*). The number of SH3-dependent PPIs of an SH3-protein is negatively correlated with the growth score of the SH3-deleted strain relative to its WT strain (Pearson's correlation = −0.426, p value = 0.038). These results highlight the critical role of SH3 domains and their PPIs in protein function.

**SH3s are rarely sufficient to dictate the breadth of PPIs driven by their host.** Having determined that multiple PPIs require SH3s, we asked whether the latter establish PPI specificity independently from their host proteins by swapping domains (Fig. 2A). Using the second round of genome editing[17], we individually replaced Abp1 SH3 with the 27 yeast SH3s and 3 human Abp1 orthologs' SH3s (CTTN, HCLS1, and DBNL, Fig. 2B). The reintroduction of its own SH3 (Abp1$_{SH3}$ in Abp1) reconstitutes almost perfectly Abp1's interaction profile (Kendall's $\tau$ = 0.93, Fig. 2B). For most cases where Abp1 loses many of its PPIs, Abp1 expression level was not significantly affected by SH3 swapping (Supplementary Fig. 2A, B). A notable exception is for instance Bem1$_{SH3-2}$ in Abp1, which leads to the loss of a large number of interactions and shows reduced abundance. This suggests a complex interplay between SH3 domains and their host proteins. No homologous SH3 domain fully re-establishes the normal Abp1 PPI profile (Fig. 2B and Supplementary Fig. 2C). However, we observe a significant correlation (cophenetic correlation = 0.20, p value = 0.005, Fig. 2C) between the similarity of PPI profiles (Fig. 2B) and the sequence similarity of the SH3s. For instance, Abp1 swapped with SH3s from its human orthologs, which have the highest sequence identity (human orthologs 46–52%, other yeast SH3s 19–41%), displays PPI profiles most strongly correlated with WT Abp1 (Kendall's $\tau$: CTTN = 0.89, DBNL = 0.89, and HCLS1 = 0.86, Fig. 2B and Supplementary Fig. 2C). Nonetheless, a subset of SH3-dependent PPIs is only observed with the endogenous domain. For instance, Hua2 and App1, both well-characterized partners of Abp1[14,18,19], are only detected when Abp1 contains its own SH3 (WT or swapped via editing). A given SH3 may therefore not be fully replaceable with other paralogous or orthologous domains. This pattern is confirmed by cellular growth phenotypic analyses (Supplementary Fig. 2D, E). PPIs, SH3 sequence similarity, and growth profiles under stress conditions are correlated (Supplementary Fig. 2D, E). This relationship is particularly clear when analyzing growth on media supplemented with hygromycin. Indeed, *ABP1* deletion

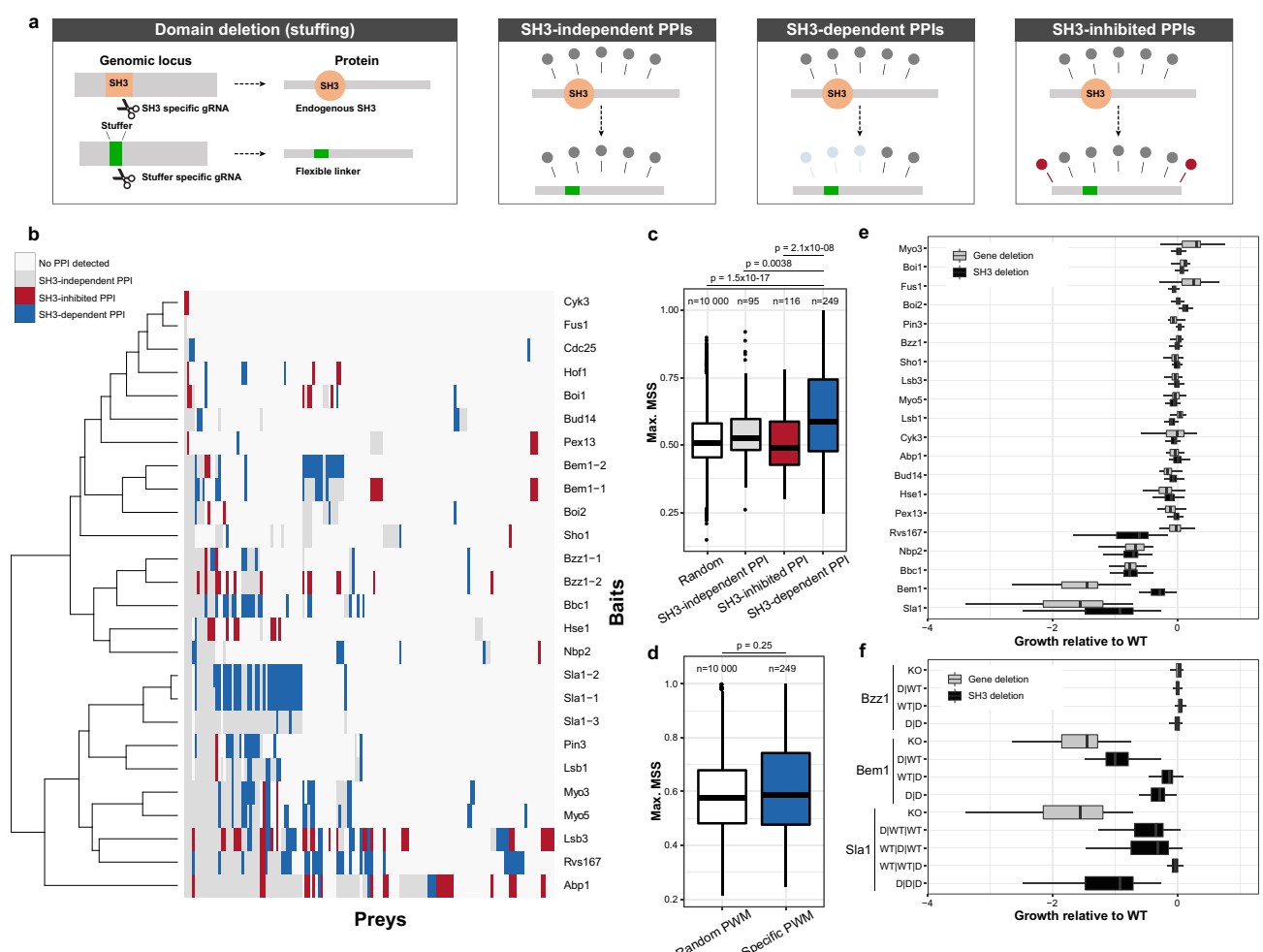

**Fig. 1 Definition of SH3-dependent PPIs in vivo and their functional impact. a** CRISPR–Cas9 SH3 editing approaches to study SH3 domains in living cells. For domain deletion (stuffing), SH3 sequences are replaced by a stuffer sequence encoding a linker using CRISPR-Cas9. SH3 dependency can then be tested. **b** PPIs of WT and SH3-deleted proteins. Colors represent different types of PPIs (assessed in quadruplicate). **c** Matrix similarity scores (MSS) of SH3-specific position weight matrices (PWMs) for prey corresponding to different types of PPIs from **b**. SH3-dependent PPIs are enriched for SH3-binding motifs relative to the Random, SH3-independent, and SH3-inhibited PPIs ($p = 1.5 \times 10^{-17}$, $p = 0.0038$ and $p = 2.1 \times 10^{-08}$, Mann–Whitney test, one-tailed, $n = 10,000$ random protein sequences, $n = 95$ preys for SH3-independent PPIs, $n = 116$ preys for SH3-inhibited PPIs, $n = 249$ preys for SH3-dependent PPIs). For example, the proportion of high MSS SH3-binding motifs (95th percentile of the MSS distribution for random peptides) determined is 11/95 for SH3-independent, 5/116 for SH3-inhibited, and 75/249 for SH3-dependent partners. **d** PWM MSS for sequences of SH3-dependent preys (as in 1C), and for PWMs randomly assigned to SH3s ($p = 0.25$, Mann–Whitney test, two-tailed, $n = 10,000$ random assignments of an SH3 PWM to an SH3 prey, $n = 249$ preys for SH3-dependent PPIs). **e** Growth of gene- or SH3-deleted strains compared to WT under 51 stress conditions. For Bzz1, Bem1, and Sla1, the data shows deletion of all SH3s. **f** Same as panel **e**, but for individual or combinations of SH3 deletions. D represents SH3 replaced with a linker sequence. Strains were grown in 12 replicates (**e**, **f** analysis performed with $n = 12$ independent colony growth per strain). For every boxplot, the median is represented as a bold center line and hinges are for the 25th and 75th percentiles (first and third quartiles). Whiskers extend from the hinges to maximum 1.5 times the Q3–Q1 interquartile range. For **c**, **d**, outliers are represented as black dots. Source data are provided as a Source Data file. See also Supplementary Data 1 and Supplementary Data 2.

leads to hygromycin resistance[20] and this phenotype is dependent on Abp1 SH3 (Supplementary Fig. 2E). None of the swapped SH3s, except for the Abp1$_{SH3}$ in Abp1, fully reproduces WT sensitivity, confirming our observation with PPI patterns (Fig. 2B and Supplementary Fig. 2E). However, the orthologous human SH3s CTTN$_{SH3}$, HCLS1$_{SH3}$, and DBNL$_{SH3}$ in Abp1 display intermediate phenotypes; this is consistent with PPI profiles and sequence similarity clusters (Fig. 2B, C and Supplementary Fig. 2E).

Several SH3 swappings lead to an inhibition of >50% of Abp1 PPIs, including SH3-independent interactions (Fig. 2B and Supplementary Fig. 2C). This suggests that SH3s can affect binding that is mediated by other regions of the protein, most likely through allosteric effects. In addition, two-thirds (21/30)

of SH3 swaps lead to gains of PPIs that were not detected with WT Abp1 (Fig. 2B and Supplementary Fig. 2C). As expected, some SH3s can bring a subset of their SH3-dependent PPIs to the Abp1 protein context; for example, Sho1$_{SH3}$ and Nbp2$_{SH3}$ in Abp1 promote the interaction with Pbs2[21–23] (Fig. 2D and Supplementary Fig. 2F). However, the majority of the gained PPIs were not identified in our SH3 deletion screen as being SH3-dependent (Fig. 2D). Thus, most SH3s are not sufficient to establish their endogenous specificity into a new protein context (Supplementary Fig. 2F). Consistent with these results, partners gained by Abp1 SH3-swapped proteins are not enriched for SH3-specific binding motifs relative to unaffected Abp1 PPIs ($p = 0.52$, Mann–Whitney test, one-sided, Supplementary Fig. 2G).

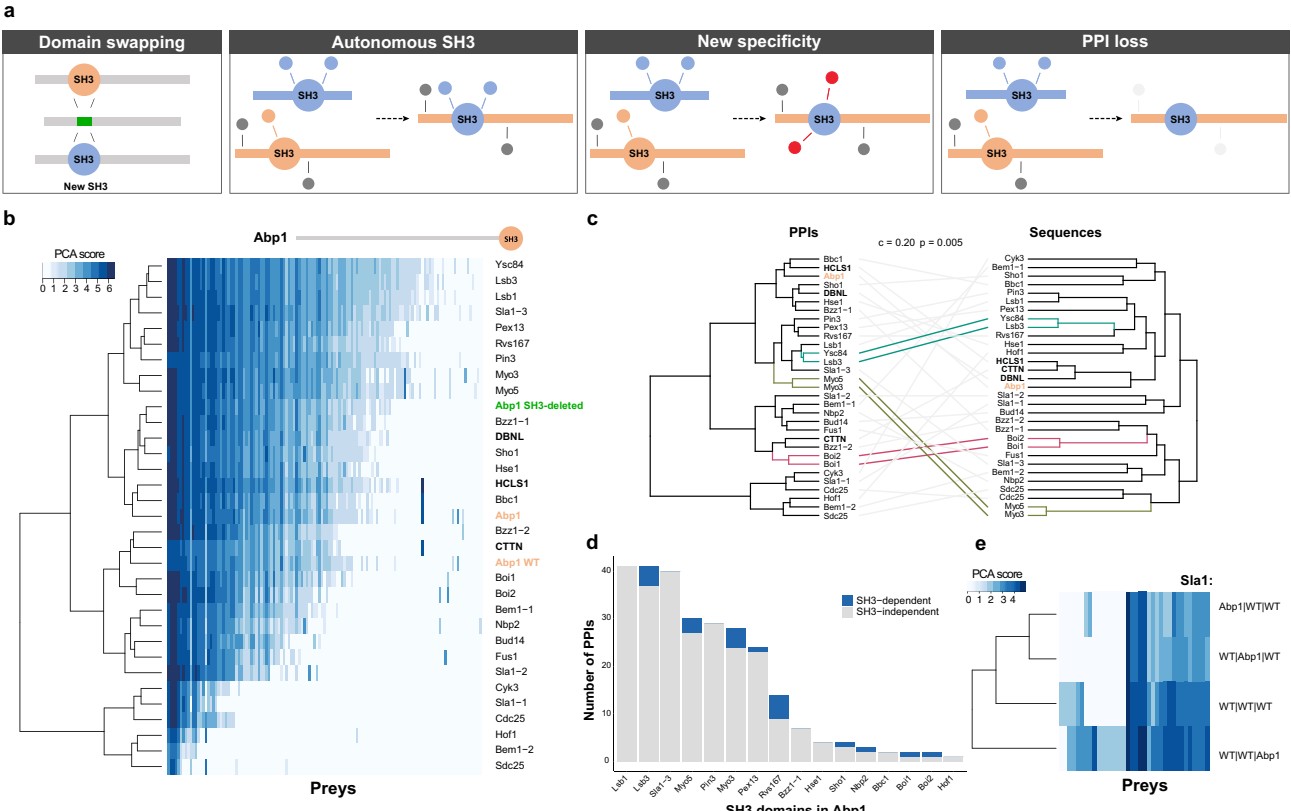

**Fig. 2 SH3 domains are rarely sufficient to establish their endogenous specificity in their host protein. a** In vivo SH3 domain swapping. The stuffer DNA from an SH3-deleted strain is replaced by SH3s from other genes. The consequences of SH3 swapping on the host protein PPIs are illustrated. **b** PPIs of Abp1 with its endogenous SH3 domain or with an SH3 from other yeast proteins or from its human orthologs. Blue shades correspond to PPI interaction scores measured by DHFR-PCA (PCA score). A scaled cartoon representation of Abp1 is shown above the heatmap. Bold SH3 domains represent human orthologous, orange SH3s are controls, and green is for the SH3-deleted protein. Each PPI was assessed in quadruplicate. **c** Cophenetic correlation between Abp1 SH3 swapped interaction clusters (based on the results of the panel **b**, $n = 4$ independent DHFR-PCA colony growth per Bait–Prey pairs) with the SH3 domain sequence similarity clusters. The empirical $p$ value ($p = 0.005$) was obtained from permutation. **d** Number of PPIs gained by Abp1 upon domain swapping. PPIs originally detected as dependent on the domain when present in its host protein are shown in blue (analysis performed with $n = 4$ independent DHFR-PCA colony growth per Bait–Prey pairs). **e** Sla1 PPIs with the SH3 from Abp1 inserted at each of the three SH3 positions. Each SH3 is either the WT Sla1 domain (WT|WT|WT) or is Abp1 SH3 (Abp1|WT|WT is Sla1 with its SH3-1 swapped with Abp1 SH3). PPIs were measured in quadruplicate. Source data are provided as a Source Data file. See also Supplementary Data 1.

We also examined whether the ability of a given SH3 to mediate PPIs depends on its position within the same host protein. We individually swapped Abp1 SH3 into each of Sla1's three SH3 positions (Fig. 2E). Inserting Abp1 SH3 at either of the first two positions only slightly affects Sla1 PPIs (Fig. 2E). SH3 swapping at the third position led to the detection of seven new PPIs despite our observation that only two Sla1 partners depend on this third SH3 (Fig. 1B, E). None of the PPIs gained by Sla1 following Abp1 SH3 swapping were originally found to be dependent on Abp1 SH3, further supporting our finding that the ability of a domain to dictate its host PPI partners is highly dependent on the identity of the host and the position of the SH3. Overall, the observation that SH3s are rarely dictating their host PPI partners by themselves, but rather alter PPIs in a manner that cannot be predicted from their intrinsic specificity suggests the presence of complex interactions between an SH3 domain and its host protein.

**Allosteric effects of SH3 domains on SH3-independent interactions**. The above analysis revealed that domain swapping impacts PPIs in a sequence-dependent manner, with divergent SH3 domains having the strongest effects. Swapping SH3s into a

protein also affects PPIs that were not previously found to be SH3-dependent, suggesting that SH3s can alter PPIs allosterically. To systematically investigate the distinction between sequence-dependent effects on SH3-dependent and SH3-independent PPIs, we measured the binding of Abp1 to an SH3-independent (Lsb3) and to an SH3-dependent partner (Hua2) for all possible single mutants of Abp1 SH3[14,19] (Fig. 3A–D and Supplementary Fig. 3A–E). We validated that mutating the Abp1 SH3 domain has little effect on its abundance (Supplementary Fig. 3C, D). Mutation sensitivity profiles for the two targets are overall highly correlated (Kendall's $\tau = 0.59$, $p$ value $= 1.0 \times 10^{-201}$), but also show significant differences (Fig. 3B–D). These results are reproducible on a small scale (Supplementary Fig. 3F), and confirm that SH3-independent PPIs (e.g., Abp1-Lsb3) are affected by changing the sequence of Abp1 SH3 (Fig. 3C).

Some positions are sensitive to any mutation for both SH3-dependent and SH3-independent PPIs (e.g., L17), while others are specific to only one (e.g., D32). Mutations affecting both PPI types correspond to buried residues that are distal from the Abp1-bound peptide and likely affect protein folding rather than amino acid-amino acid interactions[24] (Fig. 3D–F and Supplementary Fig. 3G). Positions that are only destabilizing Abp1-Hua2 lie at the peptide interface, which is in clear contrast with the sensitive

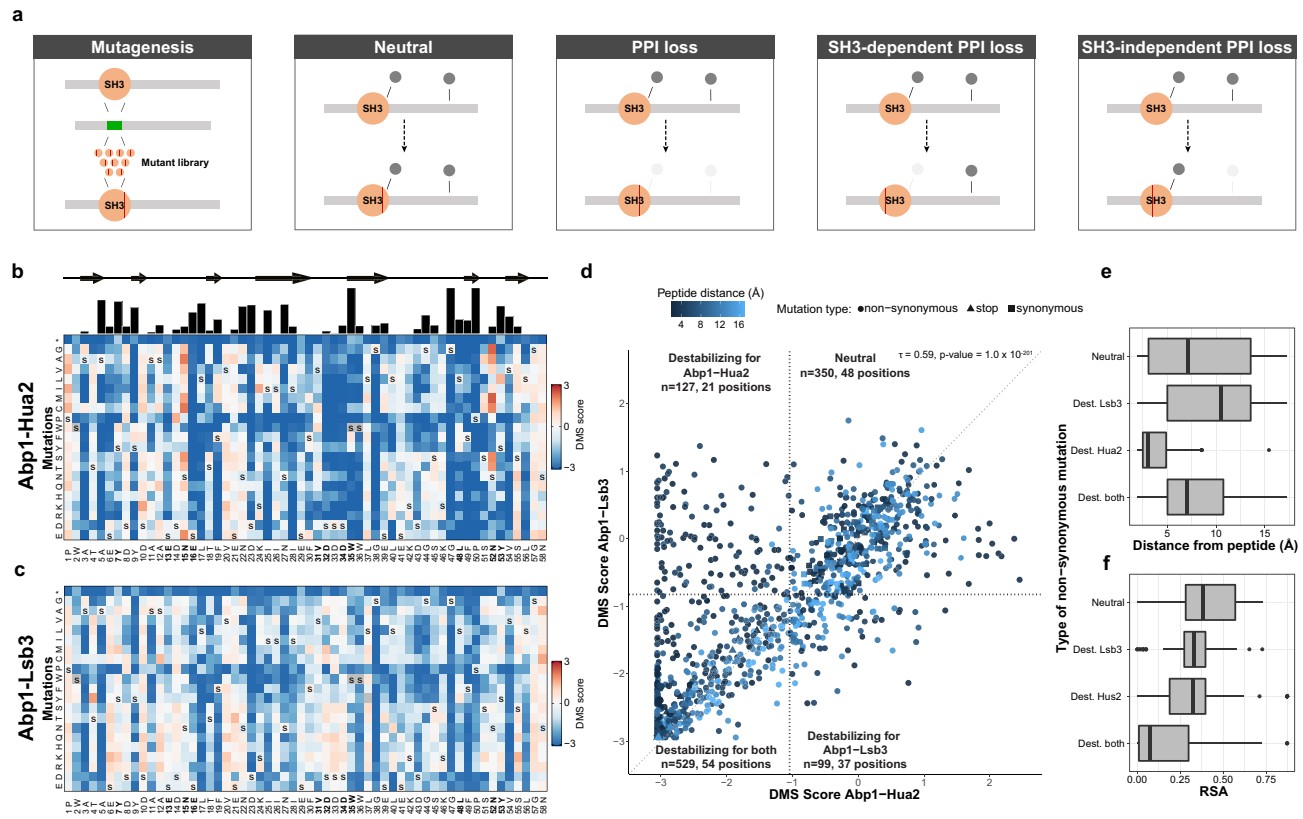

**Fig. 3 SH3 domain sequences affect both SH3-dependent and -independent PPIs. a** The stuffer DNA of an SH3-deleted strain is replaced by a library of single mutants. Categories of mutations based on their effect on PPI types are shown. **b**, **c** Impact of mutations on binding represented in terms of deep mutational scanning (DMS) score. **b** Tolerance to SH3 mutations of the SH3-dependent interaction Abp1-Hua2. Blue: low DMS score, i.e., reduced interaction strength, red: increased strength. The secondary structure of the SH3 is shown above. The black bars on the top represent the level of conservation across the 27 *S. cerevisiae* SH3s. Positions in bold represent residues in contact with the ligand (see also Supplementary Fig. 3H). **c** Same as panel **b**, but for the SH3-independent Abp1-Lsb3 PPI. The data shows the average of two biological replicates. **d** DMS scores for the two PPIs. DMS scores below the first percentile of the synonymous variant (dashed line) are defined as destabilizing the PPI. Four categories of mutants are shown: (1) destabilizing both PPIs, (2) specifically destabilizing Hua2 PPI or (3) Lsb3 PPI, and (4) neutral or slightly increasing binding. The minimum distance relative to a binding peptide is shown in blue (Ark1 peptide[24]). **e** Minimum distance to the Abp1-bound peptide and (**f**) the relative solvent accessibility (RSA) for each category. Dest is for destabilizing mutations. For **e**, **f**, *n* = 349 neutral amino acid mutations, *n* = 98 amino acid mutations that destabilize Lsb3 only, *n* = 126 amino acid mutations that destabilize Hua2 only, *n* = 528 amino acid mutations that destabilize both Lsb3 and Hua2. For the boxplots, the median is represented as a bold center line and hinges are for the 25th and 75th percentiles (first and third quartiles). Whiskers extend from the hinges to a maximum of 1.5 times the Q3–Q1 interquartile range and outliers are represented as black dots. Source data are provided as a Source Data file.

positions that specifically destabilize Abp1-Lsb3 that are distant from the SH3 binding peptide (Fig. 3D–F and Supplementary Fig. 3H). Few positions specifically affecting the Abp1-Hua2 SH3-dependent interaction are conserved between Abp1 SH3 and other yeast SH3s; this could explain why none of the other SH3s can complement its loss upon domain swapping (Figs. 2B and 3B). A subset of mutations at positions 15 and 52, both predicted to be in contact with a target peptide, specifically strengthen the Abp1-Hua2 interaction, and could in principle lead to a higher affinity (Fig. 3B, C and Supplementary Fig. 3H).

Overall, this analysis helps discriminate the residues defining SH3 binding specificity from the positions regulating the interplay between the domain and its host. The latter most likely interact in an allosteric way with the binding of partners that do not depend on the intrinsic binding specificity of the domain.

**SH3 positions are not interchangeable in multi-SH3 proteins**. Proteins containing multiple SH3s can mediate the formation of multivalent interactions, bringing an additional level of complexity to PPI regulation. The position of SH3s in their host is

highly conserved and is generally independent of the extent of their amino acid sequence conservation (Pearson's correlation = 0.22, *p* value = 0.28, Supplementary Fig. 4A–C), suggesting that SH3 positioning is key to function. To quantify the importance at the network level of SH3 position in their host, we focused on Sla1, a cytoskeleton binding protein that has three SH3s with low sequence identity (Fig. 4A, B and Supplementary Fig. 4A).

Significant differences in PPIs and growth phenotypes are dependent on Sla1 SH3 domains (Fig. 1B, F). We constructed all possible domain-position permutations within Sla1 (i.e., domain shuffling, Fig. 4A). Sla1 PPIs are highly dependent on SH3 positions (Fig. 4B and Supplementary Fig. 5A, B), which weakly correlates with Sla1 level of expression (Supplementary Fig. 5C, D). As Sla1 SH3-1 and SH3-2 bind to the same peptide motifs in vitro[14], we expected little impact from exchanging their position if peptide recognition was the sole determinant of SH3 specificity in vivo. Surprisingly, shuffling the first two domains (2|1|3) results in the loss of ~80% of Sla1 PPIs (Supplementary Fig. 5A). Many PPIs are lost when the SH3-1 position is occupied by SH3-2 (2|2|3) but maintained with SH3-3 (3|2|3), even if there are few PPIs that depend on SH3-3 when its

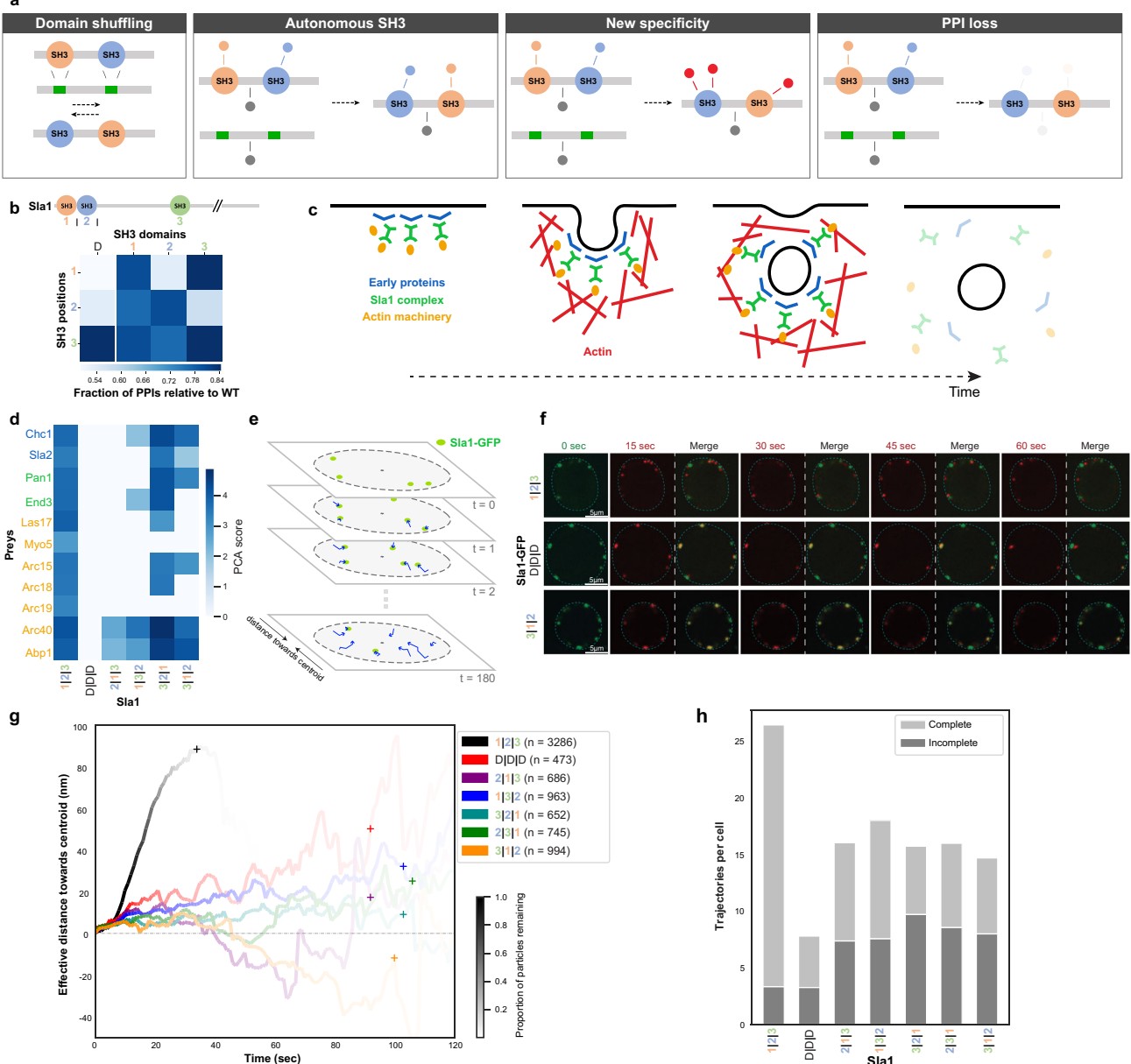

**Fig. 4 Shuffling SH3 positions alters Sla1 interactome in vivo and impacts its function in endocytosis. a** Domains are moved from one position to another in proteins containing multiple SH3s (domain shuffling). Possible outcomes on PPIs are presented. **b** Number of PPIs detected relative to WT Sla1 for SH3-shuffling or -deletion per SH3 position (measured in quadruplicates). **c** Function of Sla1 as an adaptor linking early proteins with the actin machinery in clathrin-mediated endocytosis. **d** PPIs of Sla1 SH3-shuffled or -deleted with clathrin-mediated endocytosis-related partners. The color code represents the strength of the PPIs (PCA score). Sla1 1|2|3 represents the WT protein. **e** Schematic of Sla1 foci assembled at the cell membrane and their movement toward the center of the cell during internalization before disassembly, in time. **f** Representative microscopy images of cells expressing different Sla1-GFP proteins at multiple time points. Foci from WT Sla1-GFP, the negative control D|D|D, and a Sla1-GFP shuffle (3|1|2) are shown. Green: first time frame; red: others. A merge with the starting point for each time frame is shown. **g** Average effective distance traveled by Sla1-GFP particles towards the centroid of the cell for the different proteins. Color transparency represents the proportion of events that are not completed yet. Plus signs (+) represent the moments in time when 95% of the foci have disassembled. **h** Average number of complete or incomplete events, i.e., that were not completed before the end of the image acquisition period. For panels **f–h**, the numbers of Sla1-GFP foci that were analyzed are $n = 3286$ for Sla1 1|2|3, $n = 473$ for Sla1 D|D|D, $n = 686$ for Sla1 2|1|3, $n = 963$ for Sla1 1|3|2, $n = 652$ for Sla1 3|2|1, $n = 745$ for Sla1 2|3|1 and $n = 994$ Sla1 3|1|2. Source data are provided as a Source Data file. See also Supplementary Data 1.

two first endogenous SH3 are present (Figs. 1B and 4B). Shuffling two or all three Sla1 SH3s leads to the loss of most of the Sla1 PPIs, despite all three SH3s still being present albeit in a different order (Supplementary Fig. 5A). In general, the similarity of PPI profiles of the mutants is significantly correlated with their growth phenotypes (cophenetic correlation = 0.28, $p$ value = 0.00002, from permutation, Supplementary Fig. 5B).

During clathrin-mediated endocytosis, Sla1 is an adaptor linking cargos to clathrin and recruits members of the actin machinery via SH3-dependent PPIs (Fig. 4C)[25,26]. Sla1 compartmentalizes into foci at internalization sites on the plasma membrane where it is present until vesicles are fully internalized (Fig. 4C); its deletion was shown to alter endocytosis dynamics[25–27]. The shuffling of Sla1 SH3s alters PPIs with

partners involved in different phases of endocytosis (Fig. 4D). We followed mutant Sla1 alleles by measuring the distance traveled by Sla1-GFP-labeled vesicles from the cell periphery toward the cell center (Fig. 4E and Supplementary Fig. 5E). Sla1-GFP foci movement is mostly regulated by Sla1 two N-terminal SH3s, as previously reported (Supplementary Fig. 5F, G)[26,28]. Shuffling Sla1 SH3s in any combination results in a drastic decrease in the effective distance traveled by Sla1-positive foci, as well as their persistence in time, likely due to the inability to complete the process (Fig. 4F, G and Supplementary Fig. 5H, I). This observation is consistent with a >50% decrease in the number of Sla1-GFP-labeled particle internalizations per cell and an increase in the number of incomplete events (Fig. 4H). The linearity of Sla1-GFP foci trajectories is also drastically perturbed in all Sla1 SH3-shuffled strains, suggesting inefficient vesicle internalization (Supplementary Fig. 5H). The identity and position of each of Sla1 SH3 domains, therefore, regulate its PPIs and quantitatively impact the cell's ability to tolerate stress, and to perform dynamic processes such as endocytosis.

**Protein context influences human SH3 PPIs**. We extended to humans the study of multi-SH3 proteins, as they are more pre-valent (62/216 SH3-containing proteins) than in yeast (3/23). NCK adaptors (NCK1 and NCK2) connect growth factor receptors to the actin machinery via their single SH2 and three SH3 domains[29], which bind to similar types of peptide motifs but are known to interact with different partners[6,30] (Fig. 5A). We combined affinity purification with sequential window acquisition of all theoretical mass spectra (AP-SWATH) quantitative proteomics to test whether SH3 shuffling impacts NCK2 PPIs in human cells[31,32].

We generated a high confidence WT NCK2 interactome (56 PPIs, false discovery rate (FDR) < 1%) that significantly overlaps with previously reported PPIs (71%), displaying about a third of all reported NCK2 interactors[11]. We detect significant changes in the SH3-shuffled NCK2 complexes including the complete loss of up to 17 PPIs (30% of WT NCK2 PPIs) and the gain of up to 13 new partners (Fig. 5A). The SH3-3 position appears to be more critical as NCK2 PPI profiles cluster based on the identity of the SH3 at this location (Fig. 5A). Well-characterized SH3-dependent PPIs of NCK2 depend on the position of each target cognate SH3 (Fig. 5B). In particular, NCK2 SH3-2 association with PAK1[33] is significantly impaired when the domain is at the third position (Fig. 5B and Supplementary Fig. 6A, B). In contrast, NCK2 interaction with WASL, which is mediated by multiple SH3-PXXP interactions likely involving all three NCK2 SH3s[34], is only slightly altered by shuffling (Fig. 5C). SH3 shuffling also leads to longer-range disruptions of NCK2 SH2-dependent PPIs. For example, GIT1 and P130CAS/BCAR1 associations with NCK2 SH2 are significantly modulated in several SH3-shuffled mutants[35,36] (Fig. 5C and Supplementary Fig. 6C-D). These results indicate that SH3 positioning can affect PPIs that are mediated by modular domains other than SH3s. Such indirect regulation of non-SH3 domains by adjacent SH3 positioning may also explain why many SH3-independent PPIs were altered by domain shuffling and swapping in yeast experiments.

To test whether shuffling affects the intrinsic SH3 affinity to targets via a direct effect on peptide binding, we performed in vitro fluorescence polarization binding assays using purified full-length recombinant NCK2 and peptides from direct SH3 partners CD3ε (SH3-1)[37] and PAK1 (SH3-2). SH3-1 shuffling leads to modest variations for NCK2 interaction with CD3ε, with a maximum of approximately twofold decrease when NCK2 SH3-1 is moved to the third position (Fig. 5D and Supplementary Fig. 6E). Similarly, NCK2 binding to its PAK1 target peptide

decreases by approximately twofold when its SH3-2 is inserted at the SH3-3 site (Fig. 5D and Supplementary Fig. 6F). Using circular dichroism (CD) spectropolarimetry, we determined that shuffling NCK2 SH3s does not significantly alter the overall composition of its secondary structure (Supplementary Fig. 6G). These results indicate that shuffling SH3s in their host protein might only slightly affect the position and/or accessibility of their binding pocket even though changing the positions of NCK2 SH3s resulted in the complete loss of a subset of PPIs in cells (Fig. 5A). The consequences of SH3 shuffling on the NCK2 interactome are thus possibly enhanced by factors other than binding pocket availability in vivo.

**The ability of NCK2 to phase separate in vitro depends on the position of its SH3s**. The separation of phases in the cytosol is an important feature of multiple cellular processes. The ability of NCK2's close paralog NCK1 to phase separate depends on both its number of functional domains and its capacity to self-associate[38]. We examined the ability of NCK2 SH3-shuffled proteins to undergo phase separation in vitro via self-association. Remarkably, the different NCK2 SH3-shuffled mutants initiate varying levels of phase separation (Fig. 5E). The most striking differences occur when the SH3-2 is shuffled at the third position, which results in a significant approximately twofold increase in phase separation relative to WT (Fig. 5E). Phase separation is inhibited with the addition of a PAK1 peptide, presumably by competing for homotypic binding to NCK2 SH3-2 (Supplementary Fig. 7A). Based on this and previous observations on NCK1, we hypothesized that phase separation is dependent on NCK2 SH3-2 electrostatic interaction in *trans* with the first interdomain region of another NCK2 molecule[38], which would depend on salt concentration. The propensity of NCK2 SH3-shuffled proteins to phase-separate is altered by increasing NaCl concentration; constructs bearing the SH3-2 at the third position are the most resilient (Supplementary Fig. 7B). In contrast, the addition of 1,6-hexanediol, a small aliphatic alcohol thought to impact the weak hydrophobic interactions[39], disrupts all NCK2 SH3-shuffled proteins in a similar manner (Supplementary Fig. 7C). Taken together, these results demonstrate that NCK2 phase separation is highly dependent on the position of its SH3 domains. Shuffling NCK2 SH3-2 at the third position severely disrupts NCK2 PPIs in cells (Fig. 5A) while it stabilizes NCK2 condensates in vitro (Fig. 5E), suggesting that NCK2's capacity to phase separate might impact its potential to nucleate PPIs in vivo.

PPI networks dictate how signaling pathways and biological processes are physically organized[40]. In human cells, it has been estimated that around half a million binary PPIs occur[41] and that a given individual might express nearly one million distinct proteoforms[42]. It is therefore an outstanding challenge to understand how proteins find their cognate PPI partners while avoiding spurious interactions. Here, we demonstrate that most SH3s mediate specific interactions in combination with their host proteins. How SH3 domains regulate the PPIs of their host, for example via intramolecular interactions or other allosteric effects, remains to be determined and may be dependent on the SH3 domain itself and/or the host protein structure. Our results suggest that the description of protein domains as "beads on a string" does not capture their behavior in living cells and that the protein context of structured domains is also of great importance. We assert that our findings may be extended to other families of protein binding modules, and therefore contribute to a better understanding of complex signaling networks in normal and disease states. Our observations also suggest that the evolution of protein domains by gains and losses can have complex effects that will depend on the sequential ordering of the domains

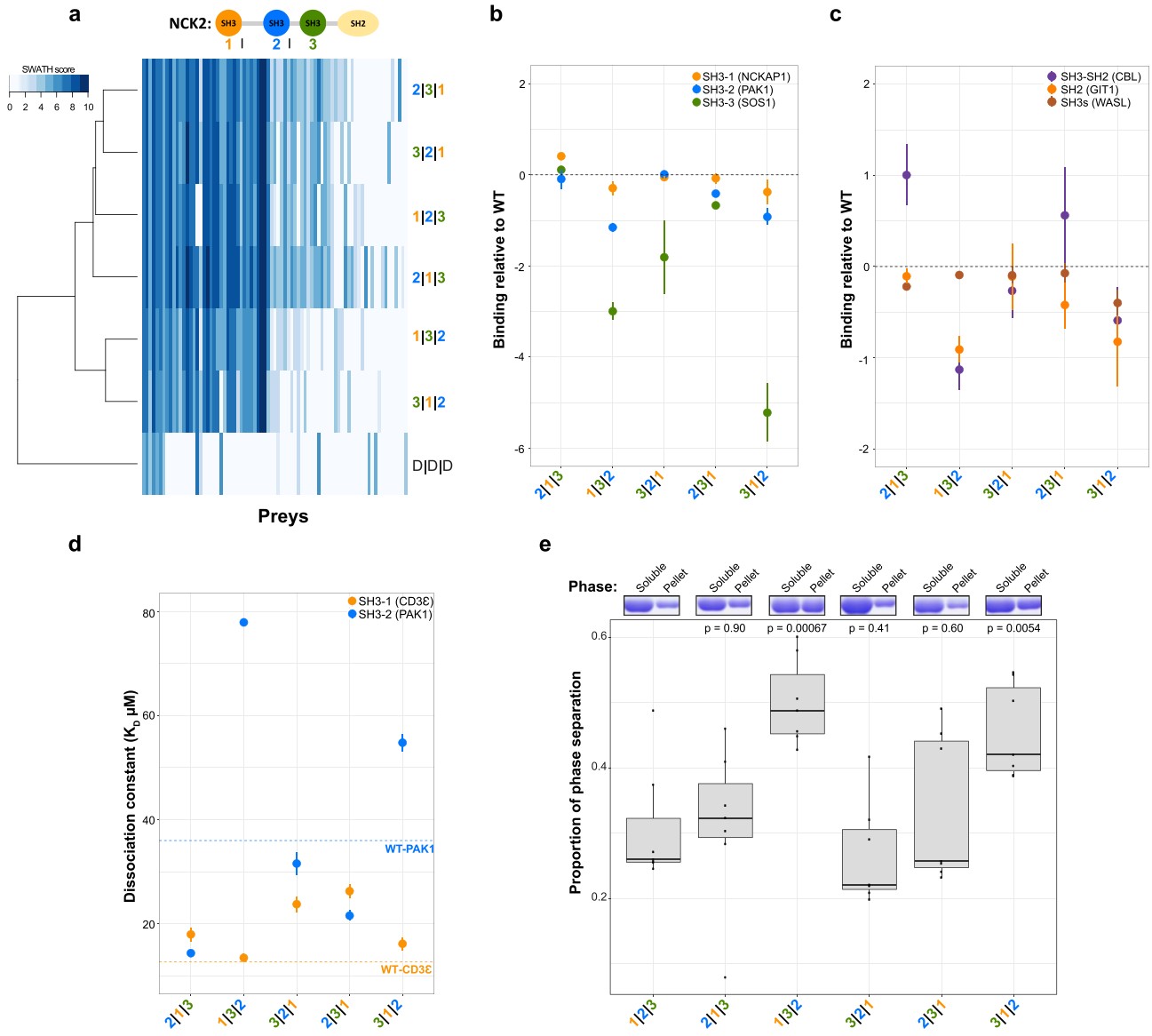

**Fig. 5 Human NCK2 SH3 domain shuffling alters its interactome in cells and its ability to phase separate. a–c** AP-SWATH quantitative MS data of NCK2 PPIs (duplicates, PPIs with a SAINT analysis FDR < 1%, see "Methods"). **a** Scaled cartoon representation of NCK2 with its domains. Log2 of the average spectral counts (SWATH score) for the PPIs are shown. Baits were clustered based on the similarity of their PPI profiles. The color code represents the PPI strength (SWATH score). The bait nomenclature 1|2|3 is for WT NCK2 and D|D|D is for the triple SH3-inactive negative control (W38K/W148K/W234K). **b**, **c** NCK2 PPIs for partners with known binding sites on NCK2. The spectral count from the two biological replicates was compared to the WT NCK2 score (Ratio NCK2mut/NCK2WT). The log2 average ratio is shown (WT NCK2 ratio = 0). Error bar represents the mean plus and minus one standard deviation ($n = 2$ independent biological experiments). **d** Fluorescence polarization dissociation constants ($K_D$) for NCK2 full-length recombinant proteins with an SH3-1 direct (CD3Ɛ) or SH3-2 direct (PAK1) partner (triplicates, $K_D$ error values represent plus and minus one SE). **e** NCK2 phase separation after 24 h of incubation. Soluble and phase-separated (pellet) proteins were quantified via Coomassie staining (typical replicate shown above). The proportion of proteins in the pellet compared to the total protein content (soluble + pellet) is shown for the seven replicates. Proportions of phase-separated proteins were compared to WT NCK2 (pairwise one-way ANOVA, p values: 2|1|3 = 0.90, 1|3|2 = 0.00067, 3|2|1 = 0.41, 2|3|1 = 0.60 and 3|1|2 = 0.0054). For the boxplot, the median is represented as a bold center line and hinges are for the 25th and 75th percentiles (first and third quartiles). Whiskers extend from the hinges to a maximum of 1.5 times the Q3–Q1 interquartile range. The black dots represent the different data points. Source data are provided as a Source Data file. See also Supplementary Data 3.

themselves, and this beyond motif recognition residues and individual domain–peptide interactions.

## Methods

***S. cerevisiae* strains**. For the DHFR-PCA experiments, bait strains were constructed and preys were either constructed or retrieved from the yeast protein interactome collection (Horizon)[43]. BY4741 (MATa, *his3Δ leu2Δ met15Δ ura3Δ*) strains with a specific gene of interest fused at its 3' to the DHFR F[1,2] with a

nourseothricin-resistance marker (NAT 100 μg/ml, Werner BioAgents) were used as baits. BY4742 (MATα, *his3Δ leu2Δ lys2Δ ura3Δ*) prey strains are each expressing a gene of interest fused at its 3' end to the DHFR F[3] and are resistant to hygromycin B (HPH 250 μg/ml, Bioshop Canada). The same BY4741 strains were used in the growth assays with the addition of knockout (KO) strains from the yeast knockout collection (Horizon)[16]. For the fluorescence microscopy experiments, the different Sla1 engineered proteins were constructed using the BY4741 Sla1-GFP strain from the GFP collection (Thermo Fisher Scientific)[44,45]. See Supplementary Table 2 for the complete list of all strains used for this study.

**S. cerevisiae growth conditions**. Cells were grown with their specific selection antibiotics in YPD medium (1% yeast extract, 2% tryptone, 2% glucose, and 2% agar (for solid medium)) and with the combination of NAT and HPH for diploid selection. Synthetic medium (0.67% yeast nitrogen base without amino acids and without ammonium sulfate, 2% glucose, 2.5% noble agar, drop-out without adenine, methionine, and lysine, and 200 µg/ml methotrexate (MTX, Bioshop Canada) diluted in dimethyl sulfoxide (DMSO, Bioshop Canada)) was used for the DHFR-PCA experiments (PCA selection). For WB experiments, cells were grown in a synthetic complete medium (0.174% yeast nitrogen base without amino acids, without ammonium sulfate, 2% glucose, 0.134% amino acid dropout complete, 0.5% ammonium sulfate). Sla1-GFP strains were grown in a synthetic complete medium without tryptophan to reduce background fluorescence. Cells were grown in a synthetic complete medium with monosodium glutamate (0.1% MSG) instead of ammonium sulfate in hygromycin B growth experiments. See Supplementary Table 3 for all the growth media used.

**Cells**. Human embryonic kidney 293T cells (HEK293T, ATCC:CRL-3216) were grown in Dulbecco's Modified Eagle's medium (DMEM) high glucose complemented with 10% fetal calf serum under 5% $CO_2$ at 37 °C. Transient expression of cDNA in cells was performed by polyethylenimine (PEI, Millipore Sigma) transfections. Enrichment of phospho-dependent PPIs was done by treating HEK293T cells with the tyrosine phosphatase inhibitor pervanadate, freshly made by mixing 4 mM orthovanadate (Millipore Sigma) with 30% $H_2O_2$ (Millipore Sigma) in water in a 200:1 volume ratio.

**Strain construction**. The complete list of strains, primers, and gRNAs used in this study can be found in Supplementary Tables 1–3. All constructed yeast strains were validated by polymerase chain reaction (PCR) and confirmed by Sanger sequencing of the locus of interest. Bait strains for the DHFR-PCA experiments were constructed as follows. The genes of interest were fused at their genomic 3′ end by replacing the stop codon with a linker (GGGGSGGGGS), the DHFR F[1,2] followed by a smaller linker (GGGGS), a 1× FLAG tag (DYKDDDDK) and the NATMX4 resistance cassette via homologous recombination (all amplified from pAG25-DHFR[1,2]-linker-FLAG). The set of 615 prey strains (DHFR F[3] fusions in the BY4742 background) were retrieved from the Yeast Protein Interactome Collection[43] with the exception of 73 that were reconstructed (Supplementary Tables 1 and 2). Knockout strains were taken from the yeast knockout collection except for *ABP1*, *BEM1*, *BZZ1*, and *SLA1*, which were reconstructed by replacing the ORFs with the NATMX4 resistance cassette by homologous recombination in the BY4741 background.

Yeast SH3 domain genomic sequences were replaced by CRISPR-Cas9 genome editing. Each SH3 domain (as defined by the SMART database V8.0 except for Ysc84 SH3 (as defined by PROSITE), see Source Data for all SH3 sequences and Supplementary Table 1 for primers) was specifically targeted by one gRNA[46]. Yeast competent cells were co-transformed with a pCAS plasmid (Addgene plasmid 60847[47]) expressing both the gRNA of interest and *Streptococcus pyogenes* Cas9[46] and a donor DNA sequence (stuffer) with 60 bp homology arms surrounding the SH3 DNA sequence. The Stuffer DNA was either PCR amplified or digested from a plasmid. The stuffer DNA sequence (GGCGGAAGTTCTGGAGGTGGTGGT) codes for a small flexible linker (GGSSGGGG) inspired by protein linkers used in structural biology[12,13]. This type of linker is frequently used for fusing proteins without affecting the structure of the linked proteins.

The stuffing of *YSC84* SH3 rendered the strain sterile and was therefore not used for this study as the assays require crosses. For genes containing two SH3s (*BZZ1* and *BEM1*), a combination of this first stuffer with a second one (GGTGGCTCAGGAGGAGGTGGTGGA) coding for a highly similar linker (GGSGGGGG) was used, which allowed targeting each SH3 position independently. The two N-terminal domains of *SLA1*, which are separated by six nucleotides, were stuffed by a single stuffer. Cells containing pCAS were selected via the G418 resistance cassette from the plasmid on YPD plates (YPD + NAT 100 µg/ml (DHFR F[1,2] selection) + G418 200 µg/mL (Bioshop Canada)). Colonies were randomly selected and grown in YPD media without G418 to allow plasmid loss. The correct insertion of the stuffer was validated by PCR and Sanger sequencing. The SH3-swapped, -shuffled, and -mutated strains were constructed following a second CRISPR–Cas9 editing step with the same procedures. The stuffer sequences were targeted by specific gRNAs (one gRNA per stuffer) and replaced by PCR amplified specific SH3 (with 40 bp homology arms) or libraries of mutated SH3 domains for Deep Mutational Scanning (DMS) with 78 bp flanking homology regions. Validation of the SH3 domain insertions was done as we previously described except for DMS libraries (see below).

**Cloning**. The fragment of plasmid pAG25-DHFR[1,2]-linker-FLAG was synthesized by Synbio Technologies. A GGGGS sequence was added after the DHFR F[1,2] followed by one repeat of the FLAG epitope (DYKDDDDK) in the parental plasmid pAG25-DHFR[1,2][43]. The cloning of the gRNAs sequences in pCAS[46] was performed via full plasmid PCR amplification with the KAPA (Roche) polymerase using 60-mer oligonucleotides (Eurofins Genomics). The resulting PCR products were digested for 1 h at 37 °C with DpnI (NEB) and transformed in competent *Escherichia coli* MC1061 cells. Positive colonies were selected on 2YT plates (1%

yeast extract, 1.6% tryptone, 0.2% glucose, 0.5% NaCl, and 2% agar) with 100 µg/ml of kanamycin. The SH3 domain DNA sequence of *ABP1* used in the DMS experiments was cloned into pUC19 (NEB) via restriction enzyme cloning with 78 bp homology arms surrounding the SH3 DNA (from genomic DNA). The sequences of CTTN, DBNL, and HCLS1 SH3 domains (gBlocks, Integrated DNA Technologies) were cloned into pUC19 in the same way. Positive clones were selected on 2YT plates with 100 µg/ml of ampicillin. For NCK2 constructs, the mouse mNCK2 cDNA sequence was cloned via restriction enzymes in frame with an N-terminal 3×FLAG tag in pcDNA3.1(−) (ThermoFisher Scientific) for western blot (WB) experiments and into the pMSCVpuro vector (Clontech) for mass spectrometry (MS) experiments. For bacterial recombinant protein expression and purification, the cDNA was inserted in frame with a 6×His-GST TEV-cleavable tag into a modified version of pET-30b (Novagen). The NCK2 SH3 shuffled chimeras were constructed using NEBuilder HiFi DNA Assembly Cloning Kit (New England Biolabs, Inc). The different functional regions of NCK2 are based on NCBI (NP_035009.3, https://www.ncbi.nlm.nih.gov/protein/NP_035009.3) and Uniprot (O55033, https://www.uniprot.org/uniprot/O55033) definitions and defined as SH3-1 (2-59), linker 1 (60-113), SH3-2 (114-170), linker 2 (171-197), SH3-3 (198-256), Linker 3 (257-282), and SH2 (283-380). All cloning was validated by Sanger sequencing.

**PCA experiments and analysis**. The overall DHFR-PCA pipeline is based on Tarassov et al.[43] and the procedures have also been published as a visualized protocol[48]. The screens were performed with robotically manipulated pin tools (BM5-SC1, S&P Robotics Inc.). First, the preys were cherry-picked from 96-well plates to omnitrays in a 384 format on YPD + HPH solid media. Plates were then combined into the 1536 format on YPD + HPH. The array was constituted of 575 preys previously shown to interact physically with at least one SH3-containing protein as reported in BioGRID V3.4.16 along with 40 negative controls[11]. The prey array comprised a border of control strains of two rows and two columns corresponding to the PPI between LSM8-DHFR F[1,2] and CDC39-DHFR F[3]. This double border was used to remove border effects. Preys are present in duplicate at a randomized position in the 1536 array (except for YAL041W which is present four times per array). Baits were crossed with two prey plates, resulting in four independent bait–prey tests for each PPI. Baits were grown in liquid YPD + NAT and then transferred on solid YPD + NAT omnitrays. They were then replicated on YPD + NAT omnitrays in 1536 array format. Mating of DHFR F[1,2] baits with DHFR F[3] preys was done on YPD followed by incubation at 30 °C for 48 h. Two diploid selection steps were performed by replicating the plates on YPD + NAT + HPH and incubating them for 48 h at 30 °C. Pictures were taken at the end of the second diploid selection for quality control. All images were acquired with a EOS Rebel T5i camera (Canon). Finally, the diploid cells were replicated on omnitrays containing solid PCA selection media and incubated for 4 days as a first selection step in a spImager custom robotic platform (S&P Robotics Inc.). Cells were replicated for a second PCA selection step and incubated in the same manner. The last time point of the second PCA selection step was used for the data analysis as in previous studies using this method.

Colony sizes from pictures were measured with the R package "Gitter"[49]. The output was validated by manual inspection. Colonies that did not grow in the diploid selection were removed from the analysis. Diploids that were still present with at least two replicates were kept for downstream analysis. The colony sizes were log2-transformed and each plate was normalized to its background level to allow for comparison among plates. The median log2 normalized colony size was then calculated for each PPI (PCA score). A cut-off was determined based on the PCA score distribution of all PPIs and on the comparison with controls to identify true PPIs (see below). To be called as a true PPI, a pair of bait–prey needed to have the median of its replicates (PCA score) above the threshold (see below). None of the baits used in the DHFR-PCA screens were found to interact with the 40 negative control preys. The first DHFR-PCA screen included WT, SH3-deleted and Abp1 SH3 swapped strains. The plates Sla1$_{SH3-1}$ in Abp1, in which Sla1 SH3-1 was used to replace Abp1 SH3, had no detectable level of background and was assigned the measured background of Abp1 WT plates. The threshold of this screen was determined based on the distribution of the PCA score of every PPI and corresponds to the top 7.3% of all detected colonies in the DHFR-PCA experiment. PPIs above the threshold of the Abp1 WT bait are highly similar (Kendall's $r = 0.93$) to those of the Abp1$_{SH3}$ in Abp1 control bait, which suggests that the determined threshold allows the detection of true interactions. In addition, we found that the true PPIs of the WT SH3-containing baits are enriched in PPIs that were previously reported (33.2%) when compared to the PPIs that did not pass the threshold (7.7% previously reported PPIs)[11]. PPI detection was also highly reproducible between replicates (Pearson's correlation = 0.91–0.93). The vast majority of the true PPIs of this screen are from four replicates (93.6%). To determine which PPIs were weaker or stronger following SH3 domain deletions, a ratio was calculated to compare the PCA scores of the PPIs of SH3-deleted baits with the PCA score of the WT baits (PCA score ratio). The same threshold was applied to the PCA score ratio, with the top 7.3% identified as stronger PPIs and the bottom 7.3% as weaker PPIs following SH3 deletions. The same principle was applied to identify PPIs that were stronger or weaker following domain swapping in Abp1. The PCA scores of the PPIs of the Abp1 SH3-swapped baits were compared to the PPIs of the Abp1$_{SH3}$ in Abp1 bait.

Liquid DHFR-PCA low-throughput experiments were performed to validate the PPIs identified as affected by the SH3 domain deletions in the high-throughput solid DHFR-PCA screen (256 PPIs weaker or lost and 116 PPIs stronger or gained following domain deletions). Mating and the diploid selection were performed as described for the large-scale experiment, with the exception that the plates were in 384 formats. Following the second diploid selection, diploid cells were inoculated in a 96 v-shaped well plate using a robotically manipulated 96 pin tool in a synthetic complete medium with MSG at pH 6.0 with NAT and HPH. After 48 h of growth, optical density (OD) was read using a TECAN Infinite F200 Pro, and dilutions to 1 OD in sterile nanopure water were prepared. Cells were then diluted to 0.1 OD by combining 25 µl of cell suspension at OD = 1 and 225 µl of PCA selection media without agar in a 96-wells plate. Plates were incubated at 30 °C in a TECAN Infinite M Nano and OD was monitored each 15 min for 72 h.

A separate DHFR-PCA screen was performed for the Sla1 SH3 shuffling experiment, with the same analysis as previously described. Three plates had no detectable level of background, which correspond to the second plate of the three Sla1 SH3 WT re-insertion controls. They were attributed the average of the background of the three first plates of the same baits. The threshold was also determined based on the distribution of the PCA scores of all PPIs and corresponds to the top 2.75% of all colonies detected in this experiment. The PPIs above the threshold of the Sla1 WT bait are highly similar (Kendall's $r = 0.89, 0.90$, and $0.87$) to those of the WT SH3 re-insertion control baits, indicating that the threshold allows the detection of true interactions. Almost all true PPIs are from four replicates (96.2%). The Sla1 SH3 shuffled strain SH3-2|SH3-3|SH3-1 DHFR F[1,2] was not successfully constructed and was excluded from the experiment.

**Growth assays in stress conditions.** All WT, SH3-deleted, KO (except for *CDC25* which is the only essential SH3-containing gene in yeast), *ABP1* SH3-swapped, and *SLA1* SH3-shuffled strains were grown on omnitrays in 1536 formats and robotically manipulated with a pin tool (the DHFR F[1,2] bait strains were used except for the KO strains). Each strain was randomly positioned in 6 replicates per plate and two plates per condition were performed, for 12 replicates per strain in total. A border of two rows and two columns of WT strains (BY4741) around the plate was used to remove border effects. Strains were grown on YPD media for two days at 30 °C. They were then replicated on 51 different media that are described in the Supplementary Table 3 and were incubated at 37 °C in a spImager custom platform with images acquired every two hours for four days. The colony sizes from plate pictures were also measured with the R package "Gitter" and the output was manually inspected. Positions with an abnormality in colony circularity were removed. The colony sizes after 74 h of incubation at 37 °C were log2 transformed and used for the analyses. First, the differences between the two plate replicates were corrected to remove plate effects. The median colony size was calculated for each strain per plate. Strains with a difference in their median colony size of more than two between the two plate replicates were removed. Finally, the average of the median colony size was calculated for each strain per condition (growth score).

For the hygromycin B resistance assays in liquid cultures, the *ABP1* SH3 swapped strains including the *ABP1* SH3-deleted and *ABP1* KO strain were grown overnight at 30 °C as precultures in synthetic complete medium with MSG. The next day, OD was measured with a TECAN Infinite F200 Pro plate reader, and the cultures were diluted to OD of 1.0 in sterile nanopure water. The different strains were further diluted at 0.1 OD by adding 25 µl of cultures to 225 µl of synthetic complete medium with MSG or with MSG and Hygromycin B 120 µg/mL in 96 wells plate (final concentration of 108 µg/mL). Cells were incubated in a TECAN Infinite M Nano plate reader at 37 °C for 24 h and OD was measured every 15 min.

**Deep mutational scanning.** Single site mutation libraries were generated by a PCR-based saturation mutagenesis method[50] as described below. The mutagenesis was carried out on the pUC19 plasmid containing the SH3 domain sequence of *ABP1* flanked by its homology arms subsequently used in the CRISPR mediated genomic re-integration. We used oligos containing degenerate nucleotides (NNN) to carry out the mutagenesis at each codon of the domain sequence. In the first step of the two-step PCR procedure (short PCR step), an amplicon was generated with an oligo containing the degenerate codon which is positioned within the domain sequence, and another oligo lying outside Abp1 sequence in the plasmid. Short PCR step was carried out with the following settings: 5 min at 95 °C, 35 cycles: 20 s at 98 °C, 15 s at 60 °C, 30 s at 72 °C, and a final extension of 1 min at 72 °C. The amplicon generated in this step was used as a mega primer in the second step (long PCR step) to amplify the whole plasmid. Long PCR step was carried out with the following settings: 5 min at 95 °C, 22 cycles: 20 s at 98 °C, 15 s at 68 °C, 3 min 30 s at 72 °C, and a final extension of 5 min at 72 °C. The long PCR product was digested for 2 h at 37 °C with DpnI. The digestion product was then transformed into *E. coli* MC1061 competent cells. In order to obtain most of the 64 possible codons, we recovered ~1000 colonies from each plate to retrieve the plasmid single mutant libraries. Saturation mutagenesis was carried out in a codon-position-wise manner. Mutants at each position were stored separately. Library quality control was assessed by amplifying the purified plasmid DNA preparations by PCR followed by Illumina sequencing as described below. The genomic insertion of the mutant libraries (one library per amino acid position) was performed by targeting the stuffer sequence in the *ABP1* SH3-deleted strain as described in the strain

construction section. *ABP1* SH3-deleted yeast competent cells were co-transformed with pCAS containing the stuffer-specific gRNA and PCR amplified *ABP1* SH3 single position mutated libraries from the pUC19 preparations. As for the library generation, ~1000 colonies from each plate were retrieved. Glycerol stocks of yeast cells were kept for each position (58 different stocks corresponding to the 58 amino acid positions of Abp1 SH3). The stocks were also validated by Illumina sequencing (see below).

**Liquid DHFR-PCA with libraries of mutants.** Liquid DHFR-PCA experiments were performed following the generation of the *ABP1* SH3 single position mutated yeast strains and were performed in biological independent duplicates. An overnight culture was started in YPD + NAT by adding the same number of cells from each yeast mutant strain (master pool). Then, the master pool and Hua2-DHFR F[3] or Lsb3-DHFR F[3] preys strains were mixed in YPD (2:1) for mating and incubated for 8 h at 30 °C. Diploid cells were selected the first time by transferring the mixture in a YPD + NAT + HPH (OD 0.5) culture for 16 h at 30 °C. The next day, the first diploid selection was transferred in SC complete pH 6.0 + NAT + HPH at 0.5 OD and the culture was grown for 24 h at 30 °C. The validation of the yeast mutant libraries was done the next day with a fraction of the diploid cells (first DHFR-PCA experiment time point without PCA selection, reference condition S2) by amplifying *ABP1* SH3 genomic sequence followed by Illumina sequencing as described in the section below. Following the selection, diploid cells OD was read using a TECAN Infinite F200 Pro, and dilutions to 0.1 OD in PCA selection media without agar in a 50 mL tube was done. Tubes were incubated at 30 °C for 72 h (second-time point of the DHFR-PCA experiment, first PCA selection). A fraction of the cells (5U OD) was collected for Illumina sequencing. The cells from the first PCA step were grown for a second PCA selection step using the same procedures. Finally, the genomic DNA of the cells (5U OD) after two cycles of PCA selection was extracted and the surviving *ABP1* SH3 mutant strains were detected by sequencing as described below.

**Single mutant libraries sequencing.** For the validation of the plasmid single mutant libraries, the plasmid DNA was extracted from bacteria following transformation using a mini-prep plasmid extraction kit (FroggaBio). Yeast genomic DNA extraction of the library of *ABP1* SH3 mutants was performed using a standard phenol/chloroform protocol. Sequencing of yeast genomic DNA was performed at three different time points during the liquid DHFR-PCA experiments. After diploid selection of the bait–prey strains (single mutant library validation, reference condition S2) and following each of the two PCA selection rounds. For the final analysis presented in this article, the first and last time points were used. The libraries for sequencing were prepared by three successive rounds of PCR. The first PCR was performed with primers to amplify *ABP1* SH3 from saturation mutagenesis minipreps (4.5 ng of plasmid) or on genomic DNA extracted from yeast (90 ng of genomic DNA) (PCR program: 3 min at 98 °C, 20 cycles: 30 s at 98 °C, 15 s at 60 °C and 30 s at 72 °C, final elongation 1 min at 72 °C). The second PCR was performed to increase diversity in the libraries by adding Row and Column barcodes[51] for identification in a 96-well plate (PCR program: 3 min at 98 °C, 15 cycles: 30 s at 98 °C, 15 s at 60 °C and 30 s at 72 °C, final elongation 1 min at 72 °C). The first PCR served as a template for the second one (2.25 µl of 1/2500 dilution). After the second PCR, 2 µl of the products were run on a 1.5% agarose gel, and band intensity was estimated using Image Lab (BioRad Laboratories). The PCR products were mixed based on their intensity on an agarose gel to roughly have equal amounts of each in the final library. Mixed PCRs were purified on magnetic beads and quantified using a NanoDrop (ThermoFisher). The third PCR was performed on 0.0045 ng of the purified pool from the second PCR to add a plate barcode and Illumina adapters (PCR program: 3 min at 98 °C, 15 cycles: 30 s at 98 °C, 15 s at 61 °C and 35 s at 72 °C, final elongation 1 min 30 s at 72 °C). Each reaction for the third PCR was performed in triplicate, combined, and purified on magnetic beads. After purification, libraries were quantified using a NanoDrop (ThermoFisher). Equal amounts of each library were combined and sent to the Genomic Analysis Platform (IBIS, Quebec, Canada) for pair-end 300 bp sequencing on a MiSeq (Illumina).

**Analysis of DMS data.** The raw data generated from deep sequencing (fastq format) was first demultiplexed based on the barcode sequences using custom scripts. Forward and reverse reads corresponding to each sample were merged using PEAR[52]. The merged fastq reads were filtered to remove reads with an average Phred score of less than 30 and the nucleotides with Phred score of less than 30 using fastp[53]. The reads were aligned to their corresponding reference sequence using bowtie2[54]. The global alignment was carried out using the command: "bowtie2 -p 6 --end-to-end --very-sensitive --no-discordant --no-mixed -x $referencep -U $fastqp -S $samp", where $referencep is the path to the bowtie2 built reference, $fastqp is the path to the unaligned fastq file and $samp is the path to the aligned file in sam format. The aligned sam file was used to count the number of mutations using samtools[55] and pysam (https://github.com/pysam-developers/pysam). In order to normalize the counts by the depth of sequencing, the counts of the mutations were divided by the depth of sequencing at the position of the mutation. The normalized counts are referred to as frequencies. Next, the log2 ratios (pseudolog with pseudo count of 0.5) of frequencies in the test condition

(second PCA selection) were normalized to the reference condition corresponding to the selection of diploids (before the PCA selections, S2) for each PPI. The python-based code used for the analysis of the DMS data is available in the align module of "rohan" package[56]. Finally, the ratio of each mutation was scaled by subtracting the median ratio for synonymous codon substitutions ($n = 120$) for each PPI (DMS score). Variants decreasing binding were defined based on the DMS score distribution for synonymous variants at the codon level. Mutations with a DMS score below the 1st percentile of this distribution represent "deleterious" variants. Likewise, mutations improving PPIs have a DMS score above the 99th percentile of this distribution. For the structural analysis, the relative solvent accessibility (RSA) values were calculated by dividing surface accessibilities by the maximum surface accessibility for the relevant amino acid. Surface accessibilities for PDB: 2RPN (10.2210/pdb2RPN/pdb)[24] were calculated using the DSSP webserver[57], and the 20 maximum surface accessibility values (empirical) taken from ref. [58]. For the conservation analysis, the 27 yeast SH3s (excluding *SDC25*-YLL017W) were aligned using the MAFFT L-INS-i method[59], and then the per-site conservation calculated with a BLOSUM62 matrix using the *conserv()* function of the R package "bio3d".

**Cellular lysis and protein immunoprecipitation.** Precultures of yeast cells expressing baits fused with the DHFR F[1,2]-FLAG were grown overnight at 30 °C with agitation in synthetic complete medium. The next day, cells were expanded in synthetic complete medium to reach their exponential phase (OD600 of around 1.0), pelleted, and frozen at −80 °C. The equivalent of fifty OD600 of cells was resuspended in ice-cold yeast lysis buffer (10 mM Tris-HCl pH 7.4, 150 mM NaCl, 0.5 mM EDTA (Millipore Sigma), 1% Triton X-100 (Millipore Sigma), and one cOmplete Mini Protease Inhibitor Cocktail (Millipore Sigma) per 10 mL) with 425–600 M glass beads (Millipore Sigma) and disrupted at 4 °C by 10 cycles of 2 min vortexing and 2 min cooldown. The equivalent of 100 OD600 of lysed cells was used for the immunoprecipitation of the bait proteins using FLAG-M2 agarose beads (Millipore Sigma). The beads were incubated with the cell lysates for 120 min at 4 °C with a slow rotation, followed by three washes with the yeast lysis buffer.

For the experiments with HEK293T cells (1 × 10cm confluent dish for WB and 3 × 15cm for MS), cells were washed in ice-cold PBS and lysed in lysis buffer (20 mM Tris-HCl pH 7.4, 150 mM NaCl, 1 mM EDTA, 0.5% sodium deoxycholate (Millipore Sigma), 10 mM beta-glycerophosphate (Millipore Sigma), 10 mM sodium pyrophosphate (Millipore Sigma), 50 mM NaF (Millipore Sigma)). Protease inhibitors were added to the lysis buffer (P8340 (Millipore Sigma) for WB and PMSF 1 mM, aprotinin 10 μg/mL, leupeptin 10 μg/mL and pepstatin 10 μg/mL (all Millipore Sigma) for MS). Cells were scraped in the lysis buffer and lysed at 4 °C for 20 min and centrifuged for another 20 min at 20,000 *g* at 4 °C. Bait proteins from the supernatant were immunoprecipitated for 90 min at 4 °C with FLAG-M2 agarose beads (Millipore Sigma) and washed three times in the lysis buffer. Beads were then used for WB or MS experiments (see below).

**Western blotting and antibodies.** Protein extracts and affinity-purified baits from yeast and human lysed cells were migrated on 10% polyacrylamide gels. The separated proteins were then transferred to nitrocellulose membranes. Validation of correct protein loading was done with ponceau staining of the membranes. Before the primary antibody incubations, membranes were blocked in 5% milk. Protein signals were detected on an Amersham Imager 600RGB (GE Healthcare) following the incubation of the membranes with the appropriate HRP-conjugated secondary antibodies and the Clarity Western ECL Substrate (Bio-Rad). The antibodies used in this study are M2-HRP (A8592, Sigma, 1/50,000), rabbit anti-FLAG (F7425, Sigma, 1/1000), PAK (SC-881, Santa Cruz, 1/200), p130cas/BCAR1 (SC-860, Santa Cruz, 1/200), actin (Cell Signaling Technology, 1/2000), anti-mouse HRP (Cell Signaling Technology, 1/10,000) and anti-rabbit HRP (Cell Signaling Technology, 1/10,000). The quantification of the WB signals was performed with the Amersham Imager 600 analysis software (GE Healthcare).

**Live imaging of endocytosis.** Overnight cultures of Sla1-GFP cells were diluted to an OD600 of 0.15 and grown in synthetic complete medium without tryptophan until they reached an OD600 of ~0.4–0.5 at 30 °C. The cells were then seeded on concanavalin A 0.05 mg/mL (Millipore Sigma) coated 8-well glass-bottom chamber slides (Sarstedt). Image acquisition was performed using a Perkin Elmer Ultra-VIEW confocal spinning disk unit attached to a Nikon Eclipse TE2000-U inverted microscope equipped with a Plan Apochromat DIC H 100×/1.4 oil objective (Nikon), and a Hamamatsu Orca Flash 4.0 LT + camera. Imaging was done at 25 °C in an environmental chamber. The software NIS-Elements (Nikon) was used for image capture. For each field, one brightfield and a series of fluorescence (GFP) images were taken. Cells were excited with a 488 nm laser and emission was filtered with a 530/630 nm filter. GFP time laps were acquired continuously at a rate of 1 frame/sec for 3 min.

**Analysis of microscopy images.** Bright-field images were used to segment cells using YeastSpotter[60]. Cells were filtered based on circularity, solidity, and the normalized difference between minor and major axis lengths to remove poorly detected cells. Out-of-focus cells were also filtered out based on size and brightness. The centroids of the segmented cells were used to identify the locations of each cell

in the image using Scikit-Image[61]. The location of each cell was used to isolate individual cell time-frames of fluorescence (GFP) images. Each cell time frame of the fluorescence images was processed through a python-based single-particle tracking tool—Trackpy[62]. Trackpy detected the locations of Sla1-GFP positive foci in each frame and linked them together to provide particle-wise trajectories. Trajectories detected in less than ten frames were considered spurious and were filtered out. Segments of the trajectories spuriously indicating movement of the particles back to the cell membrane were trimmed. The preprocessed trajectories of the particles were then used for the calculation of distances using the spatial distance module of "SciPy"[63]. The python-based code used for the analysis of microscopy images is included in the endocytosis module of "htsimaging" package[64].

**Protein purifications.** *E. coli* BL21 (DE3) cells were transformed with mNCK2 constructs in the pET-30b vector and grown overnight at 37 °C in LB with 100 μg/ml kanamycin. The next day, cultures were expanded in LB with kanamycin until they reached their exponential growth phase (OD600 of around 1.0). Protein production was then induced by the addition of 1 mM IPTG (Bio Basic) and incubation at 16 °C overnight with agitation. Cultures were pelleted with a centrifugation step (3200*g* at 4 °C, 30 min), washed with ice-cold PBS, and pellets were kept at −80 °C. Bacteria were thawed in GST buffer (20 mM Tris-HCl pH 7.5, 0.5 M NaCl, 1 mM EDTA, and 1 mM DTT (Millipore Sigma)) with cOmplete protease inhibitors (Millipore Sigma) and disrupted with sonication. Lysates were then centrifuged at 20,000*g* for 20 min at 4 °C and the supernatant loaded to a GSTrap FF 1 mL column (GE Healthcare). Following a washing step with 10 mL GST–Buffer, the GST-Tagged proteins were eluted with 15 mM GSH (Bio-basic) and cleaved in a solution using purified recombinant His-tagged TEV protease (Thermo Fisher Scientific). Tag-free NCK2 proteins were further purified using a 1 ml HisTrap FF column (GE Healthcare) to remove free His-tagged GST and TEV as well as remaining impurities. The purified proteins were concentrated with Amicon 10 K filters (Millipore Sigma) and their concentration assessed using Coomassie and BCA assays (Thermo Fisher Scientific).

**Fluorescence polarization.** Increasing amounts of NCK2 full-length recombinant proteins in GST buffer (final concentrations of 0–150 μM for CD3ε and 0–175 μM for PAK1) were used for binding assays with a constant amount of fluorescein isothiocyanate-conjugated peptides (resuspended in 17 mM Tris-HCl pH 8, 100 mM NaCl and 0.5% Brij L23 (Millipore Sigma), peptides final concentration of 40 nM). Cd3ε (RGQNKERPPPVPNPDY) and PAK1 (DIQDKPPAPPMRNTST) peptides were used as NCK2 SH3-1 and NCK2 SH3-2 ligands, respectively. Binding assays were performed in the FP buffer (17 mM Tris-HCl pH 8 and 100 mM NaCl) and fluorescence polarization was measured on a Synergy H1 multimode plate reader (Bio Tek) at 535 nm with 485 nm excitation. Calculation of the dissociation constants was performed with a one-site total binding model and nonlinear regression in GraphPad Prism version 7.

**Phase separation.** Purified NCK2 proteins were diluted in GST buffer to identical concentrations. At 500 mM NaCl present in this buffer, all NCK2 constructs were completely soluble. Phase transition was initiated by dilution in imidazole buffer (10 mM imidazole (Bio Basic) pH 7.0, 1 mM DTT, 1 mM EGTA (Millipore Sigma), 1 mM MgCl2, 2.5% Glycerol (Bioshop Canada), 3% Dextran (Millipore Sigma)) containing various amount of NaCl so that the final concentration of NaCl would correspond to those in Supplementary Fig. 7B. For all the other experiments, the phase transition of NCK2 proteins was assessed at 60 mM NaCl. NCK2 at 8 μg/μL (40 μg in 5 μL) was added to 35 μl imidazole buffer in a plastic 96-well plate that had first been pretreated with a 0.2% bovine serum albumin solution to reduce nonspecific binding. The plate assay was then incubated overnight (24 h) at 4 °C before imaging. Phase-contrast images of each well were acquired using a Zeiss Axio Vert.A1 inverted microscope with an LD A-Plan 40×/0.55 Ph1 objective (Zeiss) in Zen Blue software (version 2.3.69.1000). After imaging, the soluble protein fraction (supernatant) was recovered for sodium dodecyl sulfate poly-acrylamide gel electrophoresis analysis. Protein droplets attached at the bottom of the well (pellet) were washed once with imidazole buffer with 60 mM NaCl and then solubilized in Laemmli buffer. The pellets to supernatant fractions were used to quantify phase transition at 60 mM NaCl (7 replicates) via Coomassie protein quantification. The involvement of weak hydrophobic interactions in NCK2 driven phase transition was assessed using 1,6-Hexanediol (Millipore Sigma) at 60 mM NaCl. Imidazole buffer containing 0%, 1%, 5%, and 10% 1,6-Hexanediol was first prepared and phase transition was initiated and analyzed as described above. Finally, the competition of SH3-2 dependent homotypic NCK2 interaction with an SH3-2 binding peptide was assessed in the presence of 5, 15, 45, 90, 150, or 300 μM PAK1 peptide (DIQDKPPAPPMRNTST) in imidazole buffer with 60 mM NaCl. The impact of PAK1 peptide addition was observed via phase contrast microscopy as described above.

**CD experiments.** Far-UV CD protein spectra (250–200 nm) were recorded at 25 °C using a JASCO J-815 spectropolarimeter in a 1-mm optical path length cuvette. Fifteen-micromolar protein samples were dissolved in 20 mM phosphate buffer (pH 7.4) containing 0.3 M NaF in a total volume of 200 μL. The

temperature was maintained by a Peltier-type JASCO CDF-426S/15 thermostatic controller. Raw spectral data were extracted using the Spectra Manager Suite (JASCO). All experiments were performed in triplicate. Mean values and standard error of the mean (SEM) for each normalized data point were plotted and analyzed using GraphPad Prism 9.0. To account for small experimental variations in protein concentration during sample preparation, molar ellipticity ($\theta$, mdeg) was normalized using the following equation:

$$\text{Normalized ellipticity fraction} = \frac{\theta - \theta_{min}}{\theta_{max} - \theta_{min}} \quad (1)$$

**Experimental design for MS experiments**. NCK2 baits purified on beads from HEK293T cells (as previously described in the methods) were washed two additional times in 20 mM Tris-HCl pH 7.4 and eluted in 50 mM phosphoric acid. Proteins were then digested with trypsin (Promega) on-beads as previously detailed[65]. Briefly, eluted proteins were concentrated on ZipTip®-SCX (Millipore Sigma), reduced with TCEP 100 mM (Millipore Sigma) and alkylated during the tryptic digestion with iodoacetamide 10 mM (Millipore Sigma). Peptides were then eluted in ammonium bicarbonate 200 mM (Millipore Sigma) and desalted on Stage Tips C18 columns[66]. The final eluted peptides were dried using an Eppendorf Vacufuge™ Concentrator. For each bait, two biological replicates were processed independently. These were analyzed alongside four negative controls corresponding to transiently expressed 3×FLAG-GFP purified from HEK293T cells (as described for NCK2 constructs). These control cell lines were grown in parallel to those expressing baits and treated in the same manner. NCK2 W38/W148/W234K triple mutant, with the tryptophan residue essential for SH3 interactions mutated in each SH3, was used as a SH3-inactive negative control for SH3-dependent interactions. The equivalent of one-third of each sample was used for the SWATH-libraries generation and another third for SWATH quantification. Data-dependent acquisition (DDA) and data-independent acquisition (DIA) analysis of each sample was performed back-to-back without washes to diminish the instrument time required to complete the analysis. To minimize carry-over issues during liquid chromatography, extensive washes were performed between each sample; and the order of sample acquisition on the mass spectrometer was also reversed for the second biological replicates to avoid systematic bias. We validated that the same quantity of baits was used for each NCK2 chimeras by quantifying NCK2 peptides in each sample (see Supplementary Data 3).

**Proteins identification by MS**. The analyses were performed at the proteomic platform of the Quebec Genomics Center. Peptide samples were separated by online reversed-phase (RP) nanoscale capillary liquid chromatography (nanoLC) and analyzed by electrospray MS (ESI MS/MS). The experiments were performed with a Dionex UltiMate 3000 nanoRSLC chromatography system (Thermo Fisher Scientific) connected to a TripleTOF 5600+ mass spectrometer (Sciex) equipped with a nanoelectrospray ion source. Peptides were trapped at 20 μl/min in loading solvent (2% acetonitrile, 0.05% TFA) on a 5 mm × 300 μm C₁₈ pepmap cartridge pre-column (Thermo Fisher Scientific) for 5 min. Then, the precolumn was switch online with a self-made 50 cm × 75 μm internal diameter separation column packed with ReproSil-Pur C₁₈-AQ 3-μm resin (Dr. Maisch HPLC) and the peptides were eluted with a linear gradient from 5 to 40% solvent B (A: 0,1% formic acid, B: 80% acetonitrile, 0.1% formic acid) in 120 min, at 300 nL/min. In DDA, the instrument method consisted of one 250 ms MS1 TOF survey scan from 400 to 1300 Da followed by twenty 100 ms MS2 candidate ion scans from 100 to 2000 Da in high sensitivity mode. Only ions with a charge of 2+ to 5+ that exceeded a threshold of 150 cps were selected for MS2, and former precursors were excluded for 20 s after one occurrence. In DIA, acquisition consisted of one 50 ms MS1 scan followed by 32 × 25 a.m.u. isolation windows covering the mass range of 400–1250 a.m.u. (cycle time of 3.25 s); an overlap of 1 Da between SWATH was preselected. The collision energy for each window was set independently as defined by CE = $0.06 \times m/z + 4$, where $m/z$ is the center of each window, with a spread of 15 eV performed linearly across the accumulation time.

**DDA MS analysis**. MS data were stored, searched, and analyzed using the ProHits laboratory information management system platform[67]. Sciex.wiff MS files were converted to mzML and mzXML using ProteoWizard (3.0.4468,[68]). The mzML and mzXML files were then searched using Mascot (v2.3.02) and Comet (v2012.02 rev.0)[69]. The spectra were searched with the RefSeq database (version 57, January 30th, 2013) acquired from NCBI against a total of 72,482 human and adenovirus sequences supplemented with "common contaminants" from the Max Planck Institute (http://141.61.102.106:8080/share.cgi?ssid=0f2gfuB) and the Global Proteome Machine (GPM; http://www.thegpm.org/crap/index.html)[70]. Charges +2, +3, and +4 were allowed and the parent mass tolerance was set at 12 ppm while the fragment bin tolerance was set at 0.6 amu. Carbamidomethylation of cysteine was set as a fixed modification. Deamidated asparagine and glutamine and oxidized methionine were allowed as variable modifications. The results from each search engine were analyzed through TPP (the Trans-Proteomic Pipeline (v4.6 OCCUPY rev 3)[71], via the iProphet pipeline[72]).

**DIA MS analysis with MSPLIT-DIA**. DIA MS data was analyzed using MSPLIT-DIA (version 1.0,[73]) implemented in ProHits 4.0[67]. To generate a sample-specific spectral library for the FLAG AP-MS dataset, peptide-spectrum matches (PSMs) from matched DDA runs (18 runs) were pooled by retaining only the spectrum with the lowest MS-GFDB (Beta version 1.0072 (6/30/2014)[74]) probability for each unique peptide sequence and precursor charge state, and a peptide-level FDR of 1% was enforced using TDA[75]. The MS-GFDB parameters were set to search for tryptic cleavages, allowing no missed cleavage sites, 1 C₁₃ atom per peptide with a mass tolerance of 50 ppm for precursors with charges of 2+ to 4+ and tolerance of ±50 ppm for fragment ions. Peptide length was limited to 8–30 amino acids. Variable modifications were deamidated asparagine and glutamine and oxidized methionine. The spectra were searched with the NCBI RefSeq database (version 57, January 30th, 2013) against a total of 36,241 human and adenovirus sequences supplemented with "common contaminants" from the Max Planck Institute (http://141.61.102.106:8080/share.cgi?ssid=0f2gfuB) and the GPM (http://www.thegpm.org/crap/index.html). Decoys were appended using the decoy library command built in to MSPLIT-DIA, with a fragment mass tolerance of ±0.05 Da. The spectral library was then used for protein identification by MSPLIT[73] with peptides identified by MSPLIT-DIA passing a 1% FDR subsequently matched to genes using ProHits 4.0[67]. The MSPLIT search parameters were as follows: parent mass tolerance of ±25 Da and fragment mass tolerance of ±50 ppm. When retention time was available within the spectral library, a cut-off of ±5 min was applied to spectral matching[73].

**MS data analysis with SAINTexpress**. SAINTexpress version 3.6.1[76] was used as a statistical tool to calculate the probability value of each potential protein–protein interaction compared to background contaminants using default parameters. The four control samples were used in uncompressed mode. Two unique peptide ions and a minimum iProphet probability of 0.95 were required for protein identification prior to SAINTexpress in DDA mode while only two unique peptides were required in DIA mode.

**Data archival**. All MS files used in this study were deposited at MassIVE (http://massive.ucsd.edu). They were assigned the identifiers MassIVE MSV000085093 (DDA files) and MSV000085092 (DIA files). The files can be accessed using the following links: ftp://MSV000085092@massive.ucsd.edu and ftp://MSV000085093@massive.ucsd.edu.

**Position and sequence conservation of orthologs**. The UniProt database was used to retrieve full-length sequences of all *S. cerevisiae* SH3 proteins except for the Sdc25 (YLL017W) pseudogene[77]. One-to-one orthologs for each protein were then retrieved from fungal species using Ensembl Compara[78]. This process was automated for each SH3 protein using the Ensembl REST API[79]. The *hmmscan* function of HMMER v3.3 was then used for the domain annotation of all orthologs using default significance thresholds of 0.01 ($E = 0.01$ and dom$E = 0.01$) and the Pfam-A HMM library[80]. Any domain belonging to the Pfam SH3 clan (CL0010) was assigned as an SH3 domain. SH3 domain positions for each orthologous sequence were taken by dividing the domain start point ("env_from") by the full sequence length. This calculation was repeated for each *S. cerevisiae* homolog, and orthologous SH3 domains were considered positionally conserved if their domain positions were within a 10% sequence length window of the corresponding *S. cerevisiae* domain.

The sequence conservation of each SH3 domain was calculated using the same set of orthologous sequences as described directly above. Domain sequences within an orthologous group were aligned together using the MAFFT L-INS-i method[59]. The pairwise similarity between domain sequences was determined from the *seqidentity()* function in the R package "bio3d"[81], and we took the mean pairwise sequence similarity between the *S. cerevisiae* sequence and all its orthologs as a measure of sequence conservation for each domain.

**Tree comparison analysis**. In Fig. 2C, the relationship between SH3 domain sequences and their PPI profiles in Abp1 was explored by constructing dendrograms for both features and then quantifying the similarity between dendrograms. For the interaction similarities (Fig. 2C, left), a Euclidean distance matrix of the PCA scores was first constructed using the *dist()* function in R, and then the distance matrix used to generate a dendrogram from the *hclust()* function in R with default parameters. For the sequence similarities (Fig. 2C, right), the SH3 domain sequences were first aligned using the MAFFT L-INS-i method[59]. A distance matrix was then constructed from the *seqidentity()* function in the R package "bio3d"[81]—by taking *distance* = 1 − *similarity*—and then hierarchical clustering performed in the same manner as for the DHFR-PCA PPIs. Side-by-side visualizations of the dendrograms were generated using the R package "dendextend"[82]. The overall similarity between dendrograms was quantified using the cophenetic correlation, which correlates the tree-wise distances for both dendrograms across all possible SH3 pairs. This was calculated using the *cor.dendlist()* function in the "dendextend" package. To test for statistical significance, we permuted the data by randomizing the assignment of SH3 domains to different DHFR-PCA PPIs profiles and then recalculated the cophenetic correlation using all steps described directly above. This was repeated 10,000 times to generate a random distribution of cophenetic correlation scores. Finally, a similar tree-based approach

was used to compare PPI profiles with growth profiles for Sla1 constructs generated from the domain shuffling experiments (Supplementary Fig. 5B). Both the DHFR-PCA interaction data and the Sla1 growth data were clustered as described above, by generating a Euclidean distance matrix and then performing hierarchical clustering.

**PWMs enrichment analysis**. All yeast PWMs were taken from a 2009 study of SH3 domain specificity[14]. The similarity between SH3 PWMs and known inter-actors was assessed using a matrix similarity score (MSS) derived from the MATCH algorithm[83]. This scoring method assigns a score of 1 to perfect sequence matches to the PWM and vice versa. For each interactor assigned to an SH3 domain, MSS scores for the corresponding PWM were calculated for all possible sequence $k$-mers ($k$ = number of PWM columns) and then the maximum is taken (Max. MSS). This procedure was repeated for all SH3 domains with an assigned PWM and then the results pooled to generate Fig. 1C, D, and Supplementary Fig. 2G. Some SH3 domains were represented by more than one PWM from ref. [14], reflecting multiple specificities[84]. In these cases, MSSs were calculated for both PWMs, and then the overall maximum score took forward for further analysis.

In Fig. 1, maximum MSS scores were calculated for sequences belonging to the "Random", "SH3-independent PPI", "SH3-inhibited PPI", and "SH3-dependent PPI" category of interactor sequences. For the "Random" category, random peptides were generated by sampling amino acids according to their background frequency in the *S. cerevisiae* proteome. For SH3-independent PPI, we included only those interactors that were found to be unaffected by *any* SH3 domain deletions given that in vitro-derived SH3 PWMs overlap strongly, which could lead to spurious MSS enrichment for SH3-independent PPIs. In Fig. 1D, maximum MSSs were also recalculated for the "SH3-dependent PPI" category of interactor sequences after randomly assigning PWMs to each SH3 domain; this procedure was repeated until 10,000 maximum MSS scores were sampled. For Supplementary Fig. 2G, the "PPI gained by Abp1" corresponds to PPIs gained by Abp1 after domain swapping, and the "PPI unaffected" corresponds to PPIs of Abp1 that do not change after domain swapping. The 'Random' category was generated using the same approach described directly above for Fig. 1.

**Comparison of interactome with literature**. To find previously reported PPIs, we parsed a recent release (v 3.5.16) of the BioGRID and searched for reports of physical interactions between baits and preys[11]. PPIs were searched in the two directions possible (A–B and B–A) in the database.

**Quantification and statistical analysis**. All the statistical details of the different experiments can be found in the figure legends, figures, and results sections.

**Reporting summary**. Further information on research design is available in the Nature Research Reporting Summary linked to this article.

## Data availability

The MS datasets generated during this study are available at MassIVE; MSV000085092 (DIA SWATH MS) [doi:10.25345/C5MH55] and MSV000085093 (DDA MS) [doi:10.25345/C5GT38]. The deep sequencing datasets generated during this study are available at NCBI as BioProjects; SAMN14752885 (Abp1-Hua2 DMS reference condition), SAMN14752886 (Abp1-Lsb3 DMS reference condition), SAMN14752887 (Abp1-Hua2 DMS DHFR-PCA condition) and SAMN14752888 (Abp1-Lsb3 DMS DHFR-PCA condition). Accession codes, unique identifiers, or web links for publicly available datasets: SMART V.8.0 (http://smart.embl-heidelberg.de/, RRID:SCR_005026), PDB (PDB:2RPN, 10.2210/pdb2RPN/pdb), BioGRID (version 3.5.16, https://thebiogrid.org/, RRID:SCR_007393), RefSeq (v. 57 (01/30/2013), NCBI, https://www.ncbi.nlm.nih.gov/refseq/), Common mass spectrometry contaminants (Max Planck Institute, http://141.61.102.106:8080/share.cgi?ssid=0f2gfuB), Global Proteome Machine (https://www.thegpm.org/crap/index.html), Ensembl Compara (https://www.ensembl.org/info/genome/compara/index.html), NCBI (NP_035009.3) and Uniprot (O55033). The large datasets with the raw data for all DHFR-PCA, growth, MS, and CD experiments are included as Supplementary Data 1–3 (Excel files). Source data are provided with this paper.

## Code availability

The code generated during this study is available at DionneEtal2020 repository (github.com/Landrylab).

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

## Acknowledgements

We thank Bisson and Landry lab members for discussions; Véronique L'Italien for expert assistance with figure artwork; N. Aubin-Horth, S. Elowe, S. Michnick, I. Gagnon-Arsenault, and C. Gagné-Thivierge for critical reading of the paper. This work was supported by a Canadian Institutes of Health Research (CIHR) Foundation grant 387697 and a HFSP grant (RGP0034/2018) to CRL, CIHR Project grant 162439 to N.B., Discovery Grant 1304616-2017 from the Natural Sciences and Engineering Research Council of Canada (NSERC) to J.P.L., by the National Institute of General Medical Sciences (NIGMS) of the National Institutes of Health (NIH) (under Award R01GM105978, to N. D.) and the Natural Sciences and Engineering Research Council of Canada (NSERC) (Discovery Grants RGPIN 2016-05557 to N.D.). CRL holds the Canada Research Chair in Cellular Systems and Synthetic Biology. N.B. holds the Canada Research Chair in Cancer Proteomics. J.P.L. is supported by a Junior 1 salary award from the Fonds de Recherche du Québec-Santé (FRQ-S). N.D. holds a Fonds de Recherche Québec-Santé (FRQS) Research Scholar Senior Career Award (281993). U.D. holds an Alexander-Graham-Bell Canada graduate scholarship and the Louis-Poirier scholarship of the Fondation du CHU de Québec-Université Laval. E.B. was supported by an NSERC graduate scholarship. R.D. is funded by an FRQ-S Postdoctoral fellowship. S.D. was supported by a MITACS globalink scholarship. P.D. is supported by a CIHR Vanier Canada Graduate Scholarship. NTHP is the recipient of an FRQS Doctoral Training Scholarship.

## Author contributions

U.D., A.K.D., F.C., R.D., N.B., and C.R.L. designed the experiments. U.D., E.B., A.K.D., R.D., S.D., F.C., G.D.G., N.T.H.P., and M.L. performed the experiments. D.B., P.D., R.D., S.D., and C.R.L. designed and performed the computational analyses. U.D., E.B., D.B., R.D., P.C.D., C.R.L., J.P.L., and N.D. analyzed the results. U.D., N.B., and C.R.L. wrote the paper with contributions from all authors. N.B. and C.R.L. supervised the research.

## Competing interests

The authors declare no competing interests.
