## [Peer Review File · Nature Communications]

REVIEWER COMMENTS

Reviewer #1 (Remarks to the Author):

Dionne et al report an extensive study addressing the role of modular binding domains in determining the specificity of protein-protein interactions (PPI), using yeast and human multi-SH3 domain proteins as model systems. To this end they have created deletions, mutations, and domain swapping via genome editing, and combined this with cellular phenotyping and proteomics studies. These experiments have been carried out in a highly professional manner, and are well described and clearly discussed in the manuscript. The key conclusion based on the large body of data presented is that despite the important contribution of SH3 domains to protein function cellular PPI networks, SH3s typically do not autonomously dictate PPI specificity. Although this is not surprising in light of the current understanding on SH3 domain biology, this study emphasizes and elucidates this concept nicely, and thereby provides a valuable contribution to this field of research.

Key points:

While this study conclusively demonstrates the importance of the position of the SH3 domains within their host proteins on PPI specificity, it provides little insight into why this is the case. While such position-dependence clearly suggests allosteric effects, it would be useful to know if these are due to intramolecular SH3 interactions, modulation of other intramolecular interactions, altered accessibility of the SH3 domain, more general effects on protein folding, or some other effects.

While the main posit of the study is that SH3 domains do not autonomously dictate PPI specificity, the intrinsic capacity of the SH3 domains under study to mediate PPI 's has in fact not been examined. Although deletions and swapping of SH3 domains help to address this issue, these experiments cannot determine the capacity of an individual SH3 domain to guide a particular PPI.

Reviewer #2 (Remarks to the Author):

NCOMMS-20-35491 "Protein context shapes the specificity of domain-peptide interactions in vivo" by Dionne et al. In this m/s the authors study the factors contributing to binding specificities in SH3 domains. The work is nicely conducted, and the methodology sound. My only concern is that there is virtually no novelty in the hypothesis and conclusions of the study, and most of them were already presented about 15 years ago. Indeed, there are plenty of studies tackling the same question, following the seminal paper by Wendell Lim 's group (Zarrinpar et al. Nature 2003) and addressing the contextual specificity of peptide-mediated interactions. I really do not have any objection to the present study, other than the lack of novelty in the issue addressed and the conclusions drawn.

Probably, the most novel part is the study of the effect of SH3 domains shuffling in the regulation of phase separation processes. It would be interesting to know whether the aminoacid substitutions identified to be important to determine binding/specificity also have an influence.

Finally, I think the title is too grandiloquent, since only SH3 domain-mediated interactions are considered in the study, and the work does not offer any prove that other domain-motif interactions will behave in a similar manner.

Reviewer #3 (Remarks to the Author):

In this manuscript, Dionne et al investigate how protein-protein interactions (PPIs) of a specific protein domain can be affected by the rest of the protein that is surrounding the domain. The authors use the

SH3 family of protein domains as a model to explore this question. They delete, mutate, and shuffle the SH3 domains to show how the PPIs of a given SH3 domain can change depending on the protein context. The authors conclude that protein context is of great importance for the interactions of a protein domain.

I think that the authors are addressing a very interesting question in this paper, and I am not aware of any similar studies. They combine a diverse set of techniques to study and mutate SH3 domains, and all data appears to be included with the manuscript. Clearly the authors have done a lot of work, and have made many interesting discoveries. However, my worry with this manuscript is that a lot of the results could be explained by (1) changes in protein stability, folding, or aggregation of the mutants, or (2) by false positive or negative hits in their assays. I am missing a few crucial controls that I think are necessary to support the authors' conclusions.

Main comments:

1. On line 39-40 the authors explain how the SH3 domains were replaced by a flexible linker. Given the specific beta-barrel structure of a SH3 domain, how is replacing this domain with a flexible linker thought to affect the proteins? How was this linker designed, in order to ensure minimal disruption to the rest of the protein?

2. In many of their results (Fig 1B, 2B, 4D, S5A, 5A) the authors show that PPIs change between WT and mutant proteins. In several cases it appears that it are mainly the weaker interactions that are lost or gained. As weaker interactions would be harder to reproduce than stronger interactions, the observed differences could just be the result of false positives and negatives in the assays. Could the authors explain what the false positive and false negative rates of their assays are, how this relates to interaction strength, and what the effect of this is on the observed differences in identified interactions? ──

3. On page 5-6 the authors describe that about one third of the PPIs are lost when they replace the SH3 domain with a flexible linker, and that SH3-deletion mutants show comparable phenotypes to gene-deletion mutants. However, these observations can be the result of lower protein levels due to decreased stability of the SH3-deletion mutant protein. In Fig S1F the authors show that for 4 mutant proteins the abundance is not changed. But as knowing whether there are changes in protein stability is crucial for interpretation of the data, I think this needs to be shown for all SH3-deletion mutants that are used. Similarly, on page 9 the authors write "For cases where Abp1 loses most of its PPIs, Abp1 expression level was not significantly affected by SH3 swapping". Looking at Fig S2A, I disagree: all mutants seem to have significantly lower expression than Abp1, sometimes several fold. I have similar concerns about Fig S6, although to a lesser extent. I suggest that the authors systematically measure the expression for all mutant protein they use (Figs 1B, 2B, 2E, 4B/4D/S5A, 5A), and quantify their blots, to make sure that the observed differences in PPIs are not the result of decreased protein stability of the mutants. As far as I understood from the paper, all proteins are already tagged anyway, so this should not be an enormous effort.

4. On page 14 the authors compare the effect of mutations in the SH3 domain on PPIs with either Hua2 or Lsb3. The latter interaction is SH3-independent, the first one is not. I was wondering if the PPI interfaces of both interactions are known? Is it known to be the SH3 domain for Hua2, and is it elsewhere for Lsb3?

Minor comments:

1. On page 5, the authors mention that SH3-dependent PPI partners are enriched for SH3-binding motifs. Could they mention what fraction of SH3-dependent PPI partners actually has a SH3 binding motif?

2. I did not understand the sentence on line 49-50 / Fig 1D: "randomly assigning SH3 binding motifs

does not significantly affect the motif enrichment observed”.

3. On page 15 the authors write “Overall, this analysis helps discriminate the residues defining SH3 binding specificity from the positions regulating the core functions of the domain”. Isn’t binding specificity a core function of the domain?

Reviewer #1 (Remarks to the Author):

Dionne et al report an extensive study addressing the role of modular binding domains in determining the specificity of protein-protein interactions (PPI), using yeast and human multi-SH3 domain proteins as model systems. To this end they have created deletions, mutations, and domain swapping via genome editing, and combined this with cellular phenotyping and proteomics studies. These experiments have been carried out in a highly professional manner, and are well described and clearly discussed in the manuscript. The key conclusion based on the large body of data presented is that despite the important contribution of SH3 domains to protein function cellular PPI networks, SH3s typically do not autonomously dictate PPI specificity. Although this is not surprising in light of the current understanding on SH3 domain biology, this study emphasizes and elucidates this concept nicely, and thereby provides a valuable contribution to this field of research.

We thank this reviewer for these comments. We appreciate that this reviewer finds that our study has been performed in a highly professional manner and that it provides a valuable contribution to this field of research.

Key points:

1- While this study conclusively demonstrates the importance of the position of the SH3 domains within their host proteins on PPI specificity, it provides little insight into why this is the case. While such position-dependence clearly suggests allosteric effects, it would be useful to know if these are due to intramolecular SH3 interactions, modulation of other intramolecular interactions, altered accessibility of the SH3 domain, more general effects on protein folding, or some other effects.

We agree with this reviewer that it would be useful to experimentally determine whether the impacts observed on PPI specificity are caused by allosteric effects and/or intramolecular interactions. The molecular mechanisms explaining the effects observed on protein interactions and cellular phenotypes are likely different for every SH3 domain and thus represent a case-by-case scenario. Therefore, this is a difficult task to perform on a large scale.

For example, it was previously proposed that NCK proteins mediate an intramolecular interaction via their SH3-2 domain (Takeuchi et al. 2010; Banjade et al. 2015). Hence, it is possible that the effects on NCK2 interactions that we detect following SH3 shufflings are in part due to an alteration of this intramolecular association. However, other mechanisms are also at play. Our *in vitro* binding assays using recombinant NCK2 proteins and SH3-interacting peptides suggest that binding pocket accessibility in SH3 domains is relatively unaffected, as NCK2 SH3 domains can still mediate interactions via their peptide-binding interface following shuffling (Fig. 5D and Supplementary Fig. 6E-F). In addition, we have performed additional circular dichroism (CD)

spectropolarimetry experiments with recombinant NCK2 proteins (Supplementary Fig. 6G, described in more detail in the response to Reviewer 3). Our CD results suggest that SH3 shuffling does not significantly alter the overall secondary structure of the chimeric proteins relative to NCK2 WT. This argues that protein folding is largely unaltered following NCK2 SH3 shuffling, suggesting the existence of another mechanism to explain the allosteric effect.

Results of CD experiments were added to Supplementary Fig. 6. We also modified the manuscript discussion to incorporate this reviewer's suggestion:

It is therefore an outstanding challenge to understand how proteins find their cognate PPI partners while avoiding spurious interactions. Here, we demonstrate that most SH3s mediate specific interactions in combination with their host proteins. How SH3 domains regulate the PPIs of their host, for example via intramolecular interactions or other allosteric effects, remains to be determined and may be dependent on the SH3 domain itself and/or the host protein structure. Our results suggest that the description of protein domains as "beads on a string" is too simplistic and that the protein context of structured domains is also of great importance.

2- While the main posit of the study is that SH3 domains do not autonomously dictate PPI specificity, the intrinsic capacity of the SH3 domains under study to mediate PPI 's has in fact not been examined. Although deletions and swapping of SH3 domains help to address this issue, these experiments cannot determine the capacity of an individual SH3 domain to guide a particular PPI.

This reviewer is correct that we did not systematically analyze the impact of specifically inhibiting canonical SH3-dependent interactions, for example via point mutations in the binding pocket of all yeast SH3s. However, the goal of our study was to determine the overall contribution of SH3 domains to PPIs in an unbiased manner, including the contributions of SH3 positions outside the usual binding pockets. The importance of our approach is highlighted in our mutagenesis experiments, which show that a number of mutations at positions outside Abp1 SH3 binding interface affected its interactions.

We removed the word "autonomous" from text (4 instances) to better reflect this.

- 1) Here, we find that most SH3s ~~do not autonomously~~ do not dictate PPI specificity independently from their host protein in vivo by combining deletion, mutation, swapping and shuffling of SH3 domains and measuring their impact on protein interactions.*
- 2) Having determined that multiple PPIs require SH3s, we asked whether the latter are autonomously capable of establishing establish PPI specificity independently from their host proteins by swapping domains (Fig. 2A).*
- 3) However, the majority of the gained PPIs were not identified in our SH3 deletion screen as being SH3-dependent (Fig. 2D). Thus, most SH3s are not sufficient to*

~~do not autonomously~~ establish their endogenous specificity into a new protein context (Supplementary Fig. 2F).

- 4) None of the PPIs gained by Sla1 following Abp1 SH3 swapping were originally found to be dependent on Abp1 SH3, further supporting our finding that the ability of a domain to dictate its host PPI partners is highly dependent on the identity of the host and the position of the SH3. Overall, the observation that SH3s are rarely ~~autonomous in~~ dictating their host PPI partners by themselves, but rather alter PPIs in a manner that cannot be predicted from their intrinsic specificity suggests the presence of complex interactions between a SH3 domain and its host protein.

Reviewer #2 (Remarks to the Author):

1- NCOMMS-20-35491 “Protein context shapes the specificity of domain-peptide interactions *in vivo*” by Dionne *et al.* In this m/s the authors study the factors contributing to binding specificities in SH3 domains. The work is nicely conducted, and the methodology sound. My only concern is that there is virtually no novelty in the hypothesis and conclusions of the study, and most of them were already presented about 15 years ago. Indeed, there are plenty of studies tackling the same question, following the seminal paper by Wendell Lim’s group (Zarrinpar *et al.* Nature 2003) and addressing the contextual specificity of peptide-mediated interactions. I really do not have any objection to the present study, other than the lack of novelty in the issue addressed and the conclusions drawn.

The authors thank this reviewer for her/his/their comments. We appreciate that this reviewer finds that our work has been nicely conducted.

We acknowledge that concepts demonstrated in our study were initially discussed in the seminal paper from Dr. Lim’s group (Zarrinpar, Park, and Lim 2003), which put strong emphasis on the protein context surrounding proline-rich motifs (i.e. the sequence of the SH3 partners). Notwithstanding, the data generated from Zarrinpar *et al.* and following papers addressed questions slightly different than ours.

The main conclusion of Zarrinpar *et al.* is that negative selection shapes the evolution of the sequence around a proline-rich motif (SH3 target sequence) and could be a strong determinant of specificity in SH3-mediated interactions. Importantly, their statement that SH3 domains carry a high amount of intrinsic specificity information differs from what we observed. This major difference can probably be explained by the fact that we analyzed all yeast SH3 domains and their interactions *in vivo* in an unbiased manner when in contrast Zarrinpar *et al.* investigated the specificity determinants of a single interaction (Sho1-Pbs2). Our results show that it is incorrect to extend the finding of the Sho1-Pbs2 to all SH3-peptide interactions. It is also important to note that many of Zarrinpar *et al.* findings regarding SH3 specificity were obtained following *in vitro* experiments with isolated SH3 domains and peptides. Our experiments are specifically aimed at overcoming the limitations of *in vitro* assays.

Our manuscript also provides an exceptional resource for future research on this topic in yeast, which has been the testbed for discovery on protein interaction networks. Our contribution therefore goes beyond its conceptual advancement.

2- Probably, the most novel part is the study of the effect of SH3 domains shuffling in the regulation of phase separation processes. It would be interesting to know whether the amino acid substitutions identified to be important to determine binding/specificity also have an influence.

We appreciate that this reviewer finds that our results highlighting the importance of SH3 domain positioning for phase separation processes *in vitro* are interesting. We are currently following-up on these observations to investigate in more detail the impact of SH3-shuffling on phase separation.

3- Finally, I think the title is too grandiloquent, since only SH3 domain-mediated interactions are considered in the study, and the work does not offer any prove that other domain-motif interactions will behave in a similar manner.

We modified the title to better reflect that our work is restricted to SH3 domains. The new title now reads: “Protein context shapes the specificity of SRC-Homology 3 (SH3) domain-mediated interactions *in vivo*”.

Reviewer #3 (Remarks to the Author):

In this manuscript, Dionne et al investigate how protein-protein interactions (PPIs) of a specific protein domain can be affected by the rest of the protein that is surrounding the domain. The authors use the SH3 family of protein domains as a model to explore this question. They delete, mutate, and shuffle the SH3 domains to show how the PPIs of a given SH3 domain can change depending on the protein context. The authors conclude that protein context is of great importance for the interactions of a protein domain.

I think that the authors are addressing a very interesting question in this paper, and I am not aware of any similar studies. They combine a diverse set of techniques to study and mutate SH3 domains, and all data appears to be included with the manuscript. Clearly the authors have done a lot of work, and have made many interesting discoveries. However, my worry with this manuscript is that a lot of the results could be explained by (1) changes in protein stability, folding, or aggregation of the mutants, or (2) by false positive or negative hits in their assays. I am missing a few crucial controls that I think are necessary to support the authors' conclusions.

First, we thank this reviewer for her/his/their comments and suggestions. We appreciate this reviewer's interest in our discoveries and the acknowledgement of the amount of work presented in the manuscript.

Main comments:

1. On line 39-40 the authors explain how the SH3 domains were replaced by a flexible linker. Given the specific beta-barrel structure of a SH3 domain, how is replacing this domain with a flexible linker thought to affect the proteins? How was this linker designed, in order to ensure minimal disruption to the rest of the protein?

We thank the reviewer for this question. We had not discussed details about the linker in the main text of the initially submitted manuscript due to space constraints.

Our objective was to replace a modular SH3 domain possessing its own autonomous structure and folding independently from its host protein, as previously proposed (Zarrinpar, Park, and Lim 2003; Pawson and Nash 2003; Jin et al. 2009; Kay 2012; Bruce J. Mayer 2015; Ivarsson and Jemth 2019), by a flexible linker frequently used to fuse protein fragments without affecting their respective structure and function(s). The selected stuffer DNA sequence (GGCGGAAGTTCTGGAGGTGGTGGT) codes for a small flexible linker (GGSSGGGG) inspired from those used in structural biology (Reddy Chichili, Kumar, and Sivaraman 2013; G. Li et al. 2016). This type of Gly-rich peptide exists as natural linkers, for example in multidomain proteins (Argos 1990; Steinert et al. 1991; Reddy Chichili, Kumar, and Sivaraman 2013) and are known to be highly flexible (Arai et al. 2001; Huang et al. 2013; G. Li et al. 2016). Protein fragments on chimeric proteins constructed via this type of artificial linker typically behave

independently of one another, and their function corresponds to the combination of the two fragments of the fusion (Deane et al. 2004; Nagi and Regan 1997).

Our linker length (8 amino acids) fits exactly within the average for such linkers (between 5 to 11 amino acids) (Reddy Chichili, Kumar, and Sivaraman 2013). Polar Ser residues were added as they favorably interact with the solvent rather than the fused proteins (Argos 1990). The combination of poly-Gly with Ser should give rotational freedom to the linker, thus allowing the rest of the protein to move freely (Tollefsen et al. 2012) and enhancing solubility and resistance to proteolysis (Robinson and Sauer 1998; Eldridge et al. 2009). It is thus generally accepted that an optimal linker that limits the effects on the protein structure and functions is a combination of poly-Gly with Ser (Reddy Chichili, Kumar, and Sivaraman 2013).

We added the following text in the results section of the revised manuscript:

We repeated the experiments with baits in which the SH3s were individually replaced with a flexible linker by genome editing (SH3 domain deletion/stuffing, Fig. 1A). The stuffer DNA sequence (GGCGGAAGTTCTGGAGGTGGTGGT) codes for a small flexible poly-Gly with Ser peptide (GGSSGGGG) that is inspired from previous experiments (Reddy Chichili, Kumar, and Sivaraman 2013; G. Li et al. 2016). About a third of the SH3-containing protein interactome is qualitatively or quantitatively SH3-dependent (171 PPIs out of 607, Fig. 1B and Supplementary Fig. 1B-C).

2. In many of their results (Fig 1B, 2B, 4D, S5A, 5A) the authors show that PPIs change between WT and mutant proteins. In several cases it appears that it are mainly the weaker interactions that are lost or gained. As weaker interactions would be harder to reproduce than stronger interactions, the observed differences could just be the result of false positives and negatives in the assays. Could the authors explain what the false positive and false negative rates of their assays are, how this relates to interaction strength, and what the effect of this is on the observed differences in identified interactions?

This reviewer is correct in his/her/their observation that the interactions affected following SH3 modifications generally display a weaker PCA score, as shown here for the SH3-dependent interactions (Appendix 1A). This result is expected considering that SH3-mediated interactions are usually of low affinity (1-200 μ M range), as SH3s are involved in signaling pathways that require dynamic and transitory interactions that are also influenced by multi-valency (Bruce J. Mayer 2015; Ivarsson and Jemth 2019; Zafra-Ruano and Luque 2012; B. J. Mayer 2001; S. S.-C. Li 2005; P. Li et al. 2012; Banjade and Rosen 2014). This observation supports the involvement of SH3 domains in these types of interactions *in vivo*.

We agree with this reviewer that weaker interactions tend to be harder to reproduce and could be possibly enriched in false-positives. However, we trust that multiple lines of evidence show that the interactions we identified with lower PCA scores represent true partners.

(1) We used very stringent criteria to determine true interactions, similarly to what was previously reported (Chrétien et al. 2018; Marchant et al. 2019).

(2) DHFR experiments were highly reproducible between all four replicates (Pearson's correlation = 0.91-0.93).

(3) We detected 607 binary interactions with our 22 WT baits out of 13 530 possible PPIs. This highlights that even interactions with low PCA scores are in the top 5% of all possible interactions in the assay.

(4) The results of our small scale, more sensitive liquid PCA assay validated interactions that are affected by SH3-deletions in our large scale PCA experiment (Supplementary Fig. 1D in the original manuscript, shown here in Appendix 1B). Indeed, we find that 90% of the interactions we identify as SH3-dependent in our large scale PCA assay are also negatively affected by the SH3-deletions in the liquid PCA experiments (growth ratio SH3-deleted/WT of less than 1.00).

(5) If the differences we see for SH3-dependent interactions, which have lower PCA score, were caused by false-positives or false-negatives, we would expect these interactions to be poorly represented in the literature. However, we observed that the interactions showing best overlap with those reported in the BioGRID database are SH3-dependent, and that these interactions were for the most part discovered using methods probing for direct protein associations (Supplementary Fig. 1B-C).

(6) Sequences of the SH3-dependent protein partners are enriched for yeast SH3 binding motifs (Fig. 1C).

Together, we argue that all these observations strongly suggest that the SH3-dependent interactions, which have generally lower PCA scores (Appendix 1A), are likely true associations.

It is difficult to estimate the proportion of false negatives in our PCA experiments, largely because there is no gold standard to compare to. The interactions that are reported in the BioGRID database have been mostly detected with very different experimental approaches, such as AP-MS and Yeast Two-Hybrid. We would not expect to retrieve all of these interactions with our *in vivo* DHFR-PCA strategy involving baits and preys that are expressed endogenously. Indeed, it has been previously determined that different proteomic approaches have a low percentage of overlap in the identification of protein interactions (Yu et al. 2008). Of the 13 530 possible interactions tested with our 22 WT baits, 1192 have been reported as physical associations in BioGRID. As expected, we detect a relatively small but nonetheless significant fraction of these interactions (202/1192 (17%), representing 33% of the interactions we detected with our 22 WT baits (202/607)). However, this group is enriched for associations that were previously detected by PCA when compared to the interactions we do not detect (Appendix 1C), again supporting their high reproducibility. This proportion of overlap is similar to what

was observed when comparing interactions detected by DHFR-PCA with different specific proteomic studies (Tarassov et al. 2008). Finally, we concede that as for any other experiments on this scale, there are definitely false positives and false negatives present in our screens. Notwithstanding, we are confident that the vast majority of our SH3-dependent interactions represent true associations *in vivo*.

3. On page 5-6 the authors describe that about one third of the PPIs are lost when they replace the SH3 domain with a flexible linker, and that SH3-deletion mutants show comparable phenotypes to gene-deletion mutants. However, these observations can be the result of lower protein levels due to decreased stability of the SH3-deletion mutant protein. In Fig S1F the authors show that for 4 mutant proteins the abundance is not changed. But as knowing whether there are changes in protein stability is crucial for interpretation of the data, I think this needs to be shown for all SH3-deletion mutants that are used. Similarly, on page 9 the authors write “For cases where Abp1 loses most of its PPIs, Abp1 expression level was not significantly affected by SH3 swapping”. Looking at Fig S2A, I disagree: all mutants seem to have significantly lower expression than Abp1, sometimes several fold. I have similar concerns about Fig S6, although to a lesser extent. I suggest that the authors systematically measure the expression for all mutant protein they use (Figs 1B, 2B, 2E, 4B/4D/S5A, 5A), and quantify their blots, to make sure that the observed differences in PPIs are not the result of decreased protein stability of the mutants. As far as I understood from the paper, all proteins are already tagged anyway, so this should not be an enormous effort.

We acknowledge that the relatively high proportion of interactions negatively affected by SH3 deletions (171/607, 28%) could indicate that the SH3-deleted bait proteins are unstable. However, we have solid evidence showing that this is mostly not the case, including new data as requested by this reviewer.

We find that most interactions are not affected by the deletions (436/607, 72%). Furthermore, each of the 22 SH3-containing proteins possess interactions that are not negatively affected by the deletion of their SH3 domains (or by SH3 swappings and shufflings in Abp1, Sla1 and NCK2). This observation strongly suggests that all proteins are still functional and stable following stuffing, as destabilized proteins would tend to lose most if not all of their PPIs.

We agree with this reviewer that validating protein stability is important, which is why we had incorporated WB experiments in the original manuscript. As all our baits are endogenously tagged, their level of expression is expected to be low and can be difficult to detect using the 1xFLAG epitope. Nonetheless, we now have WB quantification for 14/26 SH3s replaced with linkers (Supplementary Fig. 1F). Our results show that deleting SH3 domains does not affect protein abundance in a manner that it would influence our interpretations. We find that there is no correlation between the number of interactions that depend on a SH3 domain and the SH3-deleted level of expression

relative to the WT protein (Pearson's $r = -0.04$, p -value = 0.90, Supplementary Fig. 1G), thus arguing against the postulate that mutant proteins display a decreased stability.

We experimentally addressed the concern of this reviewer regarding the level of expression of Abp1 SH3 swapped baits. We quantified the expression, via WB experiments, of all 33 Abp1 baits. Even though swapping Abp1 SH3 diminishes its expression level in some cases, we find a correlation that is not significant between the relative level of expression of Abp1 SH3 swapped baits and the number of interactions that they mediate, as detected by DHFR-PCA (Pearson's $r = 0.19$, p -value = 0.31, Supplementary Fig. 2A-B). These WB results suggest that indeed, in some instances, for example Abp1 with the SH3-2 of Bem1, a decrease in protein stability could explain the loss of interactions. Based on our data, this is however clearly not a general explanation for the effects observed on PPIs following the SH3 modifications. We validated that the level of expression of selected Abp1 SH3 mutants does not correlate with their ability to interact with Hua2 or Lsb3 (DMS score). Our results highlight that deleterious mutants (low DMS scores) are not systematically less expressed than WT Abp1 (Supplementary Fig. 3C-D).

In addition, we find that altering Sla1 SH3 domains seems to diminish its level of expression. However, this is only weakly associated with its ability to interact with its partners (Pearson's $r = 0.49$, p -value = 0.18, Supplementary Fig. 5C-D). Moreover, the cellular localization of Sla1 SH3-shuffled proteins were validated by live fluorescent microscopy (GFP) as shown in the original manuscript.

We quantified NCK2 peptides from our SWATH quantitative MS experiments and found little difference in their abundance, which is once again not correlated with the number of interaction partners detected (Pearson's $r = -0.26$, p -value = 0.58, Appendix 2).

To support the protein expression (WB and MS) observations, we investigated the possible effects on protein folding and overall structure of the NCK2 recombinant proteins using circular dichroism (CD). These experiments highlight that the secondary structure of NCK2 SH3-shuffled proteins is not significantly altered relative to NCK2 WT, suggesting that changing the position of SH3 domains has minimal effects on the overall structure and stability of the proteins (Supplementary Fig. 6G).

Overall, our WB and CD results demonstrate that the effects observed on protein interactions and cellular phenotypes are not primarily caused by a decrease in the expression and/or stability of the mutant proteins.

The new WB and CD data are now included in the manuscript as referenced in the text above. These sentences were added/modified to refer to the new WB and CD data:

(i) The reintroduction of its own SH3 ($Abp1_{SH3}$ in $Abp1$) reconstitutes almost perfectly $Abp1$'s interaction profile (Kendall's $\tau = 0.93$, Fig. 2B). For most cases where $Abp1$

loses many of its PPIs, Abp1 expression level was not significantly affected by SH3 swapping (Supplementary Fig. 2A-B). A notable exception is for instance Bem1_{SH3-2} in Abp1, which leads to the loss of a large number of interactions and shows reduced abundance. This suggests a complex interplay between SH3 domains and their host proteins. No homologous SH3 domain re-establishes the normal Abp1 PPI profile (Fig. 2B and Supplementary Fig. 2C).

(ii) To systematically investigate the distinction between effects on SH3-dependent and SH3-independent PPIs, we measured binding of Abp1 to a SH3-independent (Lsb3) and to a SH3-dependent partner (Hua2) for all possible single mutants of Abp1 SH3 (Fig. 3A-D and Supplementary Fig. 3A-E). We validated that mutating Abp1 SH3 domain has little effect on its abundance (Supplementary Fig. 3C-D). Mutation sensitivity profiles for the two targets are overall highly correlated (Kendall's $\tau = 0.59$, p -value = 1.0×10^{-201}), but also show significant differences (Fig. 3B-D).

(iii) We constructed all possible domain-position permutations within Sla1 (i.e. domain shuffling, Fig. 4A). Sla1 PPIs are highly dependent on SH3 positions (Fig. 4B and Supplementary Fig. 5A-B), which weakly correlates with Sla1 level of expression (Supplementary Fig. 5C-D). As Sla1 SH3-1 and SH3-2 bind to the same peptide motifs in vitro, we expected little impact from exchanging their position if peptide recognition was the sole determinant of SH3 specificity in vivo.

(iv) Similarly, NCK2 binding to its PAK1 target peptide decreases by ~2-fold when its SH3-2 is inserted at the SH3-3 site (Fig. 5D and Supplementary Fig. 6F). Using circular dichroism (CD) spectropolarimetry, we determined that shuffling NCK2 SH3s does not significantly alter the overall composition of its secondary structure (Supplementary Fig. 6G). These results indicate that shuffling SH3s in their host protein might only slightly affect the position and/or accessibility of their binding pocket even though changing the positions of NCK2 SH3s resulted in the complete loss of a subset of PPIs in cells (Fig. 5A). The consequences of SH3 shuffling on the NCK2 interactome are thus possibly enhanced by factors other than binding pocket availability in vivo.

4. On page 14 the authors compare the effect of mutations in the SH3 domain on PPIs with either Hua2 or Lsb3. The latter interaction is SH3-independent, the first one is not. I was wondering if the PPI interfaces of both interactions are known? Is it known to be the SH3 domain for Hua2, and is it elsewhere for Lsb3?

Abp1 interaction with Hua2 was identified as SH3-dependent based on our experiments. In our Abp1 SH3 swapping screen, no other SH3 domain could rescue this association (Fig. 2B). An interaction between Abp1 and Hua2 has also been reported on multiple instances (Drees et al. 2001; Yu et al. 2008; Fazi et al. 2002; Tong et al. 2002). This interaction was determined to be mediated by Abp1 SH3 domain in two of these studies (Fazi et al. 2002; Tong et al. 2002), but the SH3 residues specifically involved were not mapped. We used Abp1 SH3 domain structure in complex with a Ark1 peptide ligand

(Stollar et al. 2009) to determine the position of our mutations relative to the SH3 binding interface. The Abp1-Lsb3 interaction was unaffected by Abp1 SH3 deletion (Fig. 2B). While this association was detected in another large scale experiment (Tonikian et al. 2009), the Abp1 interface mediating the interaction was not determined.

References have now been added to the results section of the manuscript.

Minor comments:

1. On page 5, the authors mention that SH3-dependent PPI partners are enriched for SH3-binding motifs. Could they mention what fraction of SH3-dependent PPI partners actually has a SH3 binding motif?

The SH3-dependent PPI partners have on average more SH3-binding motifs (higher number of motifs with high MSS values) when compared to the other types of interactions based on their *in vitro* binding references (Fig. 1C). By considering that the 95th percentile of the distribution of the motif scores for random peptides represent optimal motifs (representing a MSS value of 0.68, the maximum being 1), we obtained these numbers:

SH3-independent: 11/95 (11.6%) with SH3 binding motif.

SH3-inhibited: 5/116 (4.3%).

SH3-dependent: 75/249 (30.1%).

These numbers were added in the legend of Figure 1.

2. I did not understand the sentence on line 49-50 / Fig 1D: “randomly assigning SH3 binding motifs does not significantly affect the motif enrichment observed”.

For figure 1C, we looked for the SH3 specific motifs that were determined *in vitro* (Tonikian et al. 2009). For example, we surveyed Lsb3 SH3 binding motif in the amino acid sequence of its partners. In figure 1D, we randomly assigned the *in vitro* SH3 motifs among the domains. For example, Protein X SH3 *in vitro* binding motif was assigned to Protein Y. We then looked for Protein X SH3 binding motif in the amino acid sequence of each Protein Y interaction partner.

The fact that we found the same kind of enrichment as in Figure 1C indicates that SH3 binding motifs determined *in vitro* are good to identify SH3-dependent interactions but not to discriminate which partner binds to which SH3 domains *in vivo*.

This sentence was modified in the manuscript:

Original: *Interestingly, randomly assigning SH3 binding motifs does not significantly affect the motif enrichment observed (p=0.25, Mann-Whitney test, two-tailed), indicating that the SH3-specific peptide motifs determined in vitro can partially identify SH3-bound proteins but poorly discriminate among different SH3 domains in vivo (Fig. 1D).*

New: Interestingly, the enrichment of SH3 motifs among SH3-dependent PPI partners does not change significantly ($p=0.25$, Mann-Whitney test, two-tailed, Fig. 1D) when assigning an SH3 binding motif randomly to a set of interactors (for example, when testing the enrichment of Protein X preferred SH3 motif among Protein Y SH3-dependent PPI partners). This indicates that SH3 binding motifs determined in vitro can adequately identify SH3-dependent interactions but not discriminate which partner binds to which SH3 domains in vivo.

3. On page 15 the authors write “Overall, this analysis helps discriminate the residues defining SH3 binding specificity from the positions regulating the core functions of the domain”. Isn’t binding specificity a core function of the domain?

We agree with the reviewer that binding specificity can be considered a core function of the domain.

We changed text as follows:

Overall, this analysis helps discriminate the residues defining SH3 binding specificity from the positions regulating the interplay between the domain and its host.

Appendix 1: Interactions with weaker PCA scores such as the SH3-dependent PPIs are true associations. **A**, Comparison of the PCA scores of SH3-dependent or -independent PPIs determined in the large scale DHFR-PCA experiments. **B**, Validation of the PPIs altered by the deletion of yeast SH3s using low-throughput liquid DHFR-PCA (from Supplementary Fig. 1D). The ratios (SH3-deleted/WT) of the optical density of the last time points from every growth curve for each PPI are shown. A ratio lower than 1 represents an interaction that is weaker upon SH3 deletion. **C**, The proportions of PPIs previously reported in the BioGRID database (v 3.5.16) that were detected or not in our large scale DHFR-PCA experiments (for the 22 WT SH3-containing baits) are shown per method of detection.

Appendix 2: NCK2 bait proteins used in AP-SWATH MS experiments are similarly abundant. The relative level of NCK2 proteins (MS relative level of expression of 0 for NCK2 WT) in relation to the number of protein partners detected for each in the MS experiments. The total count for 11 different peptides of NCK2 (covering regions not affected by the SH3-shuffling or mutations) was compared to NCK2 WT (NCK2 mutant/WT) and transformed in Log2 values (x axis). No significant Pearson correlation was calculated between the relative level of NCK2 proteins and their number of partners ($r = -0.26$, $p\text{-value} = 0.58$).

Supplementary figures with new data and updated legends

Supplementary Fig. 1| Yeast SH3 domains, the number and types of PPI changes in response to SH3 deletion and validation. **A**, The relationships of yeast SH3 amino acid sequences. Domain position and length relative to their host proteins are illustrated. SH3s used in the phylogeny are shown in orange. Grey squares are other SH3s in the same protein. The preferred type of binding motifs from in vitro assays are indicated¹². **B**, The proportion of PPIs previously reported for the different types of PPIs affected or not by SH3 deletion¹¹. **C**, The fraction of known PPIs per method of detection is shown for the same categories of interactions as in B,¹¹. **D**, Validation of the PPIs altered by the deletion of yeast SH3s-2 using low-throughput liquid DHFR-PCA. The ratios (SH3-deleted/WT) of the optical density from the last time point of the experiment for the growth curves of each PPI are shown. A ratio higher than 1 represents an interaction that is stronger upon SH3 deletion. **E**, Examples of two SH3-dependent PPIs DHFR-PCA liquid assay. Growth curves are shown for the WT and SH3-deleted baits for Sho1-Pbs2 and Nbp2-Pbs2 PPIs. The ratios of the final optical density (SH3-deleted/WT) as represented in panel D, are also indicated. **F**, Western blot (WB) analysis of the expression level of SH3-deleted proteins that lost many PPIs upon their SH3-deletion. All baits have a C-terminal 1xFLAG tag allowing their immuno-detection. **G**, Comparison of the relative level of expression of SH3-deleted baits quantified by WB (\log_2 of SH3-deleted/WT ratio, WB relative level of expression of 0 for WT baits) and their number of SH3-dependent partners identified by DHFR-PCA (Pearson's $r = -0.04$, p -value = 0.90). See also Supplementary Table 1 and 2.

Supplementary Fig. 2| Characterization of Abp1 SH3 domain swapping. A, WB analysis of protein expression levels of all Abp1 SH3 swapped bait proteins as compared to the WT Abp1, Abp1_{SH3} in Abp1 and SH3-deleted control baits. Two bands are detected for Abp1, as previously observed⁴¹. **B**, The relative level of expression of Abp1 SH3-swapped baits quantified by WB (log₂ of Abp1 SH3 swapped/Abp1_{SH3} in Abp1 ratio, WB relative level of expression of 0 for the Abp1_{SH3} in Abp1 bait) compared to their number of PPIs as identified by DHFR-PCA (Pearson's $r = 0.19$, p -value = 0.31). **C**, Number of PPIs that were affected by Abp1 SH3 domain swapping. **D**, Growth in stress conditions for the Abp1 SH3 swapped strains. The growth values were scaled per strain (row). Blue to red represents the normalized growth per strain after 74 hours (log₂ colony size). The sequence similarity of each SH3 to Abp1 SH3 is represented in grey scale. Each strain was grown in twelve replicates. **E**, Growth of Abp1 SH3 swapped stains in liquid medium with hygromycin is represented (area under the curve (AU)). The control strains, shown in black, highlight that the resistance to the drug is dependent on Abp1 SH3. Domains are sorted by increasing sequence similarity with Abp1 from the top. Growth rates were measured in triplicates. **F**, Number of SH3-dependent PPIs that were gained by each SH3 swapped to Abp1 (as determined in Figure 1B) in comparison to the ones not gained. **G**, PWMs analysis of PPIs gained by Abp1 using the matrix similarity

scores (MSS) between the PWM and sequence. This analysis shows no enrichment of the predicted SH3 motifs in the gained PPIs relative to the unaffected Abp1 PPIs ($p = 0.52$, Mann-Whitney test, one-sided). For B, D and E, the orange SH3s are controls and the SH3-deleted protein is in green. In all panels, human SH3s are shown in bold. See also Supplementary Table 3.

Supplementary Fig. 3| Characterization and validation of DMS mutants of Abp1 SH3 domain. **A-B**, Reproducibility between biological replicates of DMS DHFR-PCA experiments. The correlations for the two interactors are shown (A, is for the reference condition and B, for the DHFR-PCA condition). The frequencies of mutants (shown on log2 scale) were compared in terms of Spearman's rank correlation coefficient (rs). Associated p-value is shown on each plot. **C**, WB comparing expression levels of selected Abp1 SH3 mutants with WT Abp1. An asterisk represents a stop codon, which inhibits translation of the 1xFLAG epitope. **D**, Quantification of the relative level of expression of Abp1 SH3 mutants by WB as in C, (log2 of Abp1 SH3 mutant/Abp1 WT ratio, WB relative level of expression of 0 for WT Abp1) compared to their DMS scores (Hua2 (left): Pearson's $r = -0.34$, p-value = 0.46, Lsb3 (right): Pearson's $r = -0.81$, p-value = 0.03). **E**, Distribution of the codon log2 average sequence counts for each type of mutation of Abp1 SH3 for the two PPIs. Syn is for synonymous. **F**, Selected non-conserved positions (E6, E13, N15 and N27) sensitivities to mutations for the Abp1-Hua2 PPI were tested in a low-throughput liquid DHFR-PCA assay. The growth of the mutants in the low-throughput experiment (average of four replicates of the total OD for each growth curve, y axis) is compared to the mutants DMS score (x axis). The growth of WT Abp1 and Abp1 SH3-deleted are represented by horizontal dash lines. **G**, The median DMS score of the 58 different Abp1 SH3 positions for both PPIs in relation to their relative solvent accessibility (RSA). **H**, Abp1 SH3 residues in close proximity (within 4 Å) to the most important binding residues of the Ark1 peptide (P+2, K-3, P-4, and L-7) are mapped on the structure of Abp1 SH3 in complex with the peptide (PDB: 2RPN22). Ark1 peptide is shown in green, oxygen atoms are in red and nitrogen atoms are in blue. The colors of Abp1 SH3 residues that are mapped on the structure represent their average DMS score for Abp1-Hua2 PPI. Most interface residues sensitive to mutations, such as Y7, E13, E16, V31, D32, D34, L48 and Y53, specifically affect Hua2 PPI. See also Supplementary Table 4.

Supplementary Fig. 5| SH3 shuffling impacts Sla1 PPIs, cells growth in stress conditions and clathrin-mediated endocytosis. **A**, Sla1 SH3-deleted or -shuffled PPIs as detected by DHFR-PCA. The color code represents PPI strength as detected by DHFR-PCA (PCA score). All PPIs were measured in quadruplicate. A is for Abp1 SH3 and * is for the reinsertion of the WT domain as control strains. A scaled cartoon of Sla1 is also illustrated. **B**, Cophenetic correlation for the similarity of Sla1 SH3-shuffled PPI clusters with the growth phenotype clusters. Empirical p-value obtained from data permutation is $p = 0.00002$. **C**, Protein expression levels of the main Sla1 SH3-shuffled proteins as compared to WT Sla1. The letter “A” stands for Abp1 SH3 inserted into Sla1. **D**, The relative level of expression of Sla1 SH3-shuffled baits quantified by WB (\log_2 of Sla1 SH3-shuffled/Sla1 WT ratio, WB relative level of expression of 0 for WT Sla1) compared to their number of PPIs as identified by DHFR-PCA (Pearson’s $r = 0.49$, p -value = 0.18). The level of expression of Sla1 proteins was normalized on the level of the loading control Actin. **E**, Representative fluorescence microscopy timeframe analysis of a yeast cell expressing WT Sla1-GFP. The calculated trajectories for every Sla1-GFP foci detected are shown. Cell membrane is delimited with a dashed line and each Sla1-GFP particle color represents its position in time. **F**, Sla1-GFP particles average effective distance travelled towards the cell center through time. Sla1 SH3-deleted strains are shown. The proportion of events that are not completed yet is represented by the color transparency of the curves (+ represent the time point when 95% of foci have disassembled). **G**, Complete or incomplete Sla1-GFP endocytosis events are shown (average per cell) for Sla1 SH3-deleted strains. **H**, Linearity of Sla1-GFP particle trajectories for each SH3 deletion or shuffling. **I**, Sla1-GFP particle lifetime (in seconds) are shown for the same strains as in F., See also Supplementary Table 5.

Supplementary Fig. 6| NCK2 SH3 shuffling affects its interactions with PAK1 and P130CAS/BCAR1 in cells but only slightly alters the availability of the SH3 binding pockets *in vitro*. **A**, Western blot analysis of NCK2 SH3-shuffled proteins interaction with PAK1. Each bait is N-terminally fused to a 3XFLAG tag. Loading corresponds to Ponceau staining of the nitrocellulose membrane. Experiment was performed in four replicates in HEK293T cells. **B**, Western blot quantification of PAK1 co-immunoprecipitation with NCK2. Average log₂ ratio relative to the WT protein is shown (NCK2mut/NCK2WT, WT NCK2 log₂ ratio = 0). Ratios from replicates were compared to WT NCK2 via pairwise ANOVA statistical test (p-values: 2|1|3 = 0.017, 1|3|2 = 0.036, 3|2|1 = 0.45, 2|3|1 = 0.74 and 3|1|2 = 0.31). Error bars indicate the SD. **C**, Western blot analysis of NCK2 SH3-shuffled proteins association with SH2 target P130CAS/BCAR1. The experiment was performed in four replicates in HEK293T cells stimulated with the tyrosine phosphatase inhibitor pervanadate. **D**, Quantification of the Western blot signals of P130CAS co-immunoprecipitation with NCK2 as in B,. The four ratios of the replicates were compared to WT NCK2 via pairwise ANOVA statistical test (p-values: 2|1|3 = 0.13, 1|3|2 = 0.0045, 3|2|1 = 0.027, 2|3|1 = 0.0076 and 3|1|2 = 0.019). Error bars indicate the SD. **E-F**, Fluorescence polarization *in vitro* curves for NCK2 binding to CD3E or PAK1. Each binding assay was executed in triplicate. Error bars indicate the SD of the average of the ΔFP for each point. Dissociation constants error values represent the SE of the value derived from the binding curve. **G**, Far-UV CD spectra of NCK2 recombinant proteins. WT NCK2 spectrum (black) is overlaid on SH3-shuffled proteins or NCK2 D|D|D triple SH3-inactive negative control (W38K/W148K/W234K) spectra. Bold traces and vertical dashed lines represent the mean and SEM calculated values for each triplicate measurement, respectively. WT NCK2, NCK2 D|D|D and NCK2 SH3-shuffled proteins similarly exhibit a mix between random coil and folded secondary structure elements. See also Supplementary Table 6.

References:

- Arai, R., H. Ueda, A. Kitayama, N. Kamiya, and T. Nagamune. 2001. "Design of the Linkers Which Effectively Separate Domains of a Bifunctional Fusion Protein." *Protein Engineering* 14 (8): 529–32.
- Argos, P. 1990. "An Investigation of Oligopeptides Linking Domains in Protein Tertiary Structures and Possible Candidates for General Gene Fusion." *Journal of Molecular Biology* 211 (4): 943–58.
- Banjade, Sudeep, and Michael K. Rosen. 2014. "Phase Transitions of Multivalent Proteins Can Promote Clustering of Membrane Receptors." *eLife* 3 (October). <https://doi.org/10.7554/eLife.04123>.
- Banjade, Sudeep, Qiong Wu, Anuradha Mittal, William B. Peeples, Rohit V. Pappu, and Michael K. Rosen. 2015. "Conserved Interdomain Linker Promotes Phase Separation of the Multivalent Adaptor Protein Nck." *Proceedings of the National Academy of Sciences of the United States of America* 112 (47): E6426–35.
- Chrétien, Andr  e-  ve, Isabelle Gagnon-Arsenault, Alexandre K. Dub  , Xavier Barbeau, Philippe C. Despr  s, Claudine Lamothe, Anne-Marie Dion-C  t  , Patrick Lag  ie, and Christian R. Landry. 2018. "Extended Linkers Improve the Detection of Protein-Protein Interactions (PPIs) by Dihydrofolate Reductase Protein-Fragment Complementation Assay (DHFR PCA) in Living Cells." *Molecular & Cellular Proteomics: MCP* 17 (2): 373–83.
- Deane, Janet E., Daniel P. Ryan, Margaret Sunde, Megan J. Maher, J. Mitchell Guss, Jane E. Visvader, and Jacqueline M. Matthews. 2004. "Tandem LIM Domains Provide Synergistic Binding in the LMO4:Ldb1 Complex." *The EMBO Journal* 23 (18): 3589–98.
- Drees, B. L., B. Sundin, E. Brazeau, J. P. Caviston, G. C. Chen, W. Guo, K. G. Kozminski, et al. 2001. "A Protein Interaction Map for Cell Polarity Development." *The Journal of Cell Biology* 154 (3): 549–71.
- Eldridge, Bill, R. Neil Cooley, Richard Odegrip, Duncan P. McGregor, Kevin J. Fitzgerald, and Christopher G. Ullman. 2009. "An in Vitro Selection Strategy for Conferring Protease Resistance to Ligand Binding Peptides." *Protein Engineering, Design & Selection: PEDS* 22 (11): 691–98.
- Fazi, Barbara, M. Jamie T. V. Cope, Alice Douangamath, Silvia Ferracuti, Katja Schirwitz, Adriana Zucconi, David G. Drubin, Matthias Wilmanns, Gianni Cesareni, and Luisa Castagnoli. 2002. "Unusual Binding Properties of the SH3 Domain of the Yeast Actin-Binding Protein Abp1: Structural and Functional Analysis." *The Journal of Biological Chemistry* 277 (7): 5290–98.
- Huang, Ziliang, Fengchun Ye, Chong Zhang, Shuo Chen, Yin Chen, Jingjun Wu, Masahiro Togo, and Xin-Hui Xing. 2013. "Rational Design of a Tripartite Fusion Protein of Heparinase I Enables One-Step Affinity Purification and Real-Time Activity Detection." *Journal of Biotechnology* 163 (1): 30–37.
- Ivarsson, Ylva, and Per Jemth. 2019. "Affinity and Specificity of Motif-Based Protein-Protein Interactions." *Current Opinion in Structural Biology* 54 (February): 26–33.
- Jin, Jing, Xueying Xie, Chen Chen, Jin Gyoon Park, Chris Stark, D. Andrew James, Marina Olhovskiy, Rune Linding, Yongyi Mao, and Tony Pawson. 2009. "Eukaryotic Protein Domains as Functional Units of Cellular Evolution." *Science Signaling* 2 (98): ra76.
- Kay, Brian K. 2012. "SH3 Domains Come of Age." *FEBS Letters* 586 (17): 2606–8.
- Li, Gang, Ziliang Huang, Chong Zhang, Bo-Jun Dong, Ruo-Hai Guo, Hong-Wei Yue, Li-Tang Yan, and Xin-Hui Xing. 2016. "Construction of a Linker Library with Widely Controllable Flexibility for Fusion Protein Design." *Applied Microbiology and*

- Biotechnology* 100 (1): 215–25.
- Li, Pilong, Sudeep Banjade, Hui-Chun Cheng, Soyeon Kim, Baoyu Chen, Liang Guo, Marc Llaguno, et al. 2012. “Phase Transitions in the Assembly of Multivalent Signalling Proteins.” *Nature* 483 (7389): 336–40.
- Li, Shawn S-C. 2005. “Specificity and Versatility of SH3 and Other Proline-Recognition Domains: Structural Basis and Implications for Cellular Signal Transduction.” *Biochemical Journal* 390 (Pt 3): 641–53.
- Marchant, Axelle, Angel F. Cisneros, Alexandre K. Dubé, Isabelle Gagnon-Arsenault, Diana Ascencio, Honey Jain, Simon Aubé, et al. 2019. “The Role of Structural Pleiotropy and Regulatory Evolution in the Retention of Heteromers of Paralogs.” *eLife* 8 (August). <https://doi.org/10.7554/eLife.46754>.
- Mayer, B. J. 2001. “SH3 Domains: Complexity in Moderation.” *Journal of Cell Science* 114 (Pt 7): 1253–63.
- Mayer, Bruce J. 2015. “The Discovery of Modular Binding Domains: Building Blocks of Cell Signalling.” *Nature Reviews. Molecular Cell Biology* 16 (11): 691–98.
- Nagi, A. D., and L. Regan. 1997. “An Inverse Correlation between Loop Length and Stability in a Four-Helix-Bundle Protein.” *Folding and Design* 2 (1): 67–75.
- Pawson, Tony, and Piers Nash. 2003. “Assembly of Cell Regulatory Systems through Protein Interaction Domains.” *Science* 300 (5618): 445–52.
- Reddy Chichili, Vishnu Priyanka, Veerendra Kumar, and J. Sivaraman. 2013. “Linkers in the Structural Biology of Protein-Protein Interactions.” *Protein Science: A Publication of the Protein Society* 22 (2): 153–67.
- Robinson, C. R., and R. T. Sauer. 1998. “Optimizing the Stability of Single-Chain Proteins by Linker Length and Composition Mutagenesis.” *Proceedings of the National Academy of Sciences of the United States of America* 95 (11): 5929–34.
- Steinert, P. M., J. W. Mack, B. P. Korge, S. Q. Gan, S. R. Haynes, and A. C. Steven. 1991. “Glycine Loops in Proteins: Their Occurrence in Certain Intermediate Filament Chains, Loricrins and Single-Stranded RNA Binding Proteins.” *International Journal of Biological Macromolecules* 13 (3): 130–39.
- Stollar, Elliott J., Bianca Garcia, P. Andrew Chong, Arianna Rath, Hong Lin, Julie D. Forman-Kay, and Alan R. Davidson. 2009. “Structural, Functional, and Bioinformatic Studies Demonstrate the Crucial Role of an Extended Peptide Binding Site for the SH3 Domain of Yeast Abp1p.” *The Journal of Biological Chemistry* 284 (39): 26918–27.
- Takeuchi, Koh, Zhen-Yu J. Sun, Sunghyoun Park, and Gerhard Wagner. 2010. “Autoinhibitory Interaction in the Multidomain Adaptor Protein Nek: Possible Roles in Improving Specificity and Functional Diversity.” *Biochemistry* 49 (27): 5634–41.
- Tarassov, Kirill, Vincent Messier, Christian R. Landry, Stevo Radinovic, Mercedes M. Serna Molina, Igor Shames, Yelena Malitskaya, Jackie Vogel, Howard Bussey, and Stephen W. Michnick. 2008. “An in Vivo Map of the Yeast Protein Interactome.” *Science* 320 (5882): 1465–70.
- Tollefsen, Stig, Kinya Hotta, Xi Chen, Bjørg Simonsen, Kunchithapadam Swaminathan, Irimpan I. Mathews, Ludvig M. Sollid, and Chu-Young Kim. 2012. “Structural and Functional Studies of Trans-Encoded HLA-DQ2.3 (DQA1*03:01/DQB1*02:01) Protein Molecule.” *The Journal of Biological Chemistry* 287 (17): 13611–19.
- Tong, Amy Hin Yan, Becky Drees, Giuliano Nardelli, Gary D. Bader, Barbara Brannetti, Luisa Castagnoli, Marie Evangelista, et al. 2002. “A Combined Experimental and Computational Strategy to Define Protein Interaction Networks for Peptide Recognition Modules.” *Science* 295 (5553): 321–24.
- Tonikian, Raffi, Xiaofeng Xin, Christopher P. Toret, David Gfeller, Christiane Landgraf, Simona Panni, Serena Paoluzi, et al. 2009. “Bayesian Modeling of the Yeast SH3

- Domain Interactome Predicts Spatiotemporal Dynamics of Endocytosis Proteins.” *PLoS Biology* 7 (10): e1000218.
- Yu, Haiyuan, Pascal Braun, Muhammed A. Yildirim, Irma Lemmens, Kavitha Venkatesan, Julie Sahalie, Tomoko Hirozane-Kishikawa, et al. 2008. “High-Quality Binary Protein Interaction Map of the Yeast Interactome Network.” *Science* 322 (5898): 104–10.
- Zafra-Ruano, Ana, and Irene Luque. 2012. “Interfacial Water Molecules in SH3 Interactions: Getting the Full Picture on Polyproline Recognition by Protein-Protein Interaction Domains.” *FEBS Letters* 586 (17): 2619–30.
- Zarrinpar, Ali, Sang-Hyun Park, and Wendell A. Lim. 2003. “Optimization of Specificity in a Cellular Protein Interaction Network by Negative Selection.” *Nature* 426 (6967): 676–80.

REVIEWERS' COMMENTS

Reviewer #1 (Remarks to the Author):

In the revised version of the manuscript by Dionne et al the authors have provided new CD data to rule out major folding defects due to SH3 domain shuffling of NCK2, added a couple of new sentences, and deleted some words ("autonomously" in particular) to partially address my criticism, which was largely agreed with in their rebuttal letter. While these changes have improved the manuscript somewhat, the weak points of the study explained in my original Key points remain the same. However, even with these shortcomings this study has significant merit and provides a large body of solid data. I continue to be pleased with the technical quality of the study and impressed with the amount of experimental work involved, but not equally enthusiastic about the originality or groundbreaking nature of the ideas and conclusions presented.

Reviewer #2 (Remarks to the Author):

As I said in my initial assessment, this is a well-conducted study, and I do not really have anything to criticize. My main (and almost only) point was its lack of novelty of both hypothesis and results. I only cited the work of Wendel Lim in my assessment, which is the one commented by the authors, but there are many studies addressing similar questions and reaching similar conclusions (i.e. from Tony Pawson and/or Mike Yaffe, to name a couple). Nevertheless, it is a nice and thorough confirming paper and, if the Editor is fine with it, I am happy to support its publication.

Reviewer #3 (Remarks to the Author):

I appreciate the authors' effort to address the concerns raised by me and other reviewers. I am satisfied with the changes the authors made to the manuscript, and I have no further comments.

Reviewer #4 (Remarks to the Author):

The manuscript from Dionne et al describes a set of experiments aimed at assessing the contribution of broader context to protein interactions classically described as domain driven using the SH3 domain as a canonical example. In my view the experiments appear very well done, the data is well described with clear illustration, and the conclusions are compelling and proportional to the data. The conclusion that SH3 domain binding specificity is not independent of the remaining context of the protein would certainly seem to what is expected intuitively (with some prior published literature as discussed by reviewer 2). However, in my view the careful empirical demonstration of this effect (including in yeast and human systems, with respect to allosteric effects, domain swapping, and very interestingly phase separation) is very compelling and certainly adds new knowledge to the literature.

As I come to this manuscript after the initial review and rebuttal has been completed I would also offer the opinion that the answers to the reviewers questions are convincing and that the authors have done a generally good job to address both the technical and conceptual questions put to them.

Further the editor asked that I would look specifically at technical aspects of the mass spectrometry driven experiments. These experiments were done using a well established and published strategy for testing quantitatively the impact of perturbations on protein-protein interactions via affinity purification and mass spectrometry (AP-SWATH). The authors apply this method essentially as previously published, the analysis appears well done, and the data is made public via a repository. I see no issues here.

I would suggest the manuscript is suitable for publication as is.

Reviewer #1 (Remarks to the Author):

In the revised version of the manuscript by Dionne et al the authors have provided new CD data to rule out major folding defects due to SH3 domain shuffling of NCK2, added a couple of new sentences, and deleted some words (“autonomously” in particular) to partially address my criticism, which was largely agreed with in their rebuttal letter. While these changes have improved the manuscript somewhat, the weak points of the study explained in my original Key points remain the same. However, even with these shortcomings this study has significant merit and provides a large body of solid data. I continue to be pleased with the technical quality of the study and impressed with the amount of experimental work involved, but not equally enthusiastic about the originality or groundbreaking nature of the ideas and conclusions presented.

Reviewer #2 (Remarks to the Author):

As I said in my initial assessment, this is a well-conducted study, and I do not really have anything to criticize. My main (and almost only) point was its lack of novelty of both hypothesis and results. I only cited the work of Wendel Lim in my assessment, which is the one commented by the authors, but there are many studies addressing similar questions and reaching similar conclusions (i.e. from Tony Pawson and/or Mike Yaffe, to name a couple). Nevertheless, it is a nice and thorough confirming paper and, if the Editor is fine with it, I am happy to support its publication.

Reviewer #3 (Remarks to the Author):

I appreciate the authors' effort to address the concerns raised by me and other reviewers. I am satisfied with the changes the authors made to the manuscript, and I have no further comments.

Reviewer #4 (Remarks to the Author):

The manuscript from Dionne et al describes a set of experiments aimed at assessing the contribution of broader context to protein interactions classically described as domain driven using the SH3 domain as a canonical example. In my view the experiments appear very well done, the data is well described with clear illustration, and the conclusions are compelling and proportional to the data. The conclusion that SH3 domain binding specificity is not independent of the remaining context of the protein would certainly seem to what is expected intuitively (with some prior published literature as discussed by reviewer 2). However, in my view the careful empirical demonstration of this effect (including in yeast and human systems, with respect to allosteric effects, domain swapping, and very interestingly phase separation) is very compelling and certainly adds new knowledge to the literature.

As I come to this manuscript after the initial review and rebuttal has been completed I would also offer the opinion that the answers to the reviewers questions are convincing and that the authors have done a generally good job to address both the technical and conceptual questions put to them.

Further the editor asked that I would look specifically at technical aspects of the mass spectrometry driven experiments. These experiments were done using a well established and published strategy for testing

quantitatively the impact of perturbations on protein-protein interactions via affinity purification and mass spectrometry (AP-SWATH). The authors apply this method essentially as previously published, the analysis appears well done, and the data is made public via a repository. I see no issues here.

I would suggest the manuscript is suitable for publication as is.

We thank all the reviewers for their comments and inputs. We greatly appreciate the time and efforts that each reviewer has put into the review of our manuscript, especially in these hard times.